# Early life gut microbiota sustains liver-resident natural killer cells maturation via the butyrate-IL-18 axis

Panpan Tian[1], Wenwen Yang[1], Xiaowei Guo ⓘ[1], Tixiao Wang[1], Siyu Tan[1], Renhui Sun[1], Rong Xiao[1], Yuzhen Wang[1], Deyan Jiao[1], Yachen Xu[1], Yanfei Wei[1], Zhuanchang Wu[1,2], Chunyang Li ⓘ[3], Lifen Gao ⓘ[1,2], Chunhong Ma ⓘ[1,2] ✉ & Xiaohong Liang ⓘ[1,2] ✉

Liver-resident natural killer cells, a unique lymphocyte subset in liver, develop locally and play multifaceted immunological roles. However, the mechanisms for the maintenance of liver-resident natural killer cell homeostasis remain unclear. Here we show that early-life antibiotic treatment blunt functional maturation of liver-resident natural killer cells even at adulthood, which is dependent on the durative microbiota dysbiosis. Mechanistically, early-life antibiotic treatment significantly decreases butyrate level in liver, and subsequently led to defective liver-resident natural killer cell maturation in a cell-extrinsic manner. Specifically, loss of butyrate impairs IL-18 production in Kupffer cells and hepatocytes through acting on the receptor GPR109A. Disrupted IL-18/IL-18R signaling in turn suppresses the mitochondrial activity and the functional maturation of liver-resident natural killer cells. Strikingly, dietary supplementation of experimentally or clinically used *Clostridium butyricum* restores the impaired liver-resident natural killer cell maturation and function induced by early-life antibiotic treatment. Our findings collectively unmask a regulatory network of gut-liver axis, highlighting the importance of the early-life microbiota in the development of tissue-resident immune cells.

The liver is a key and frontline immune organ, particularly enriched for innate immune cells including innate lymphoid cells (ILCs), natural killer T (NKT) cells, macrophages and γδ T cells[1,2]. These innate immune cells act coordinately to eliminate invading pathogens, as well as to maintain liver functional homeostasis[3]. ILCs accounts for about 5% intrahepatic lymphocytes in mice and 25% in human at steady state and majorly consists of conventional natural killer (cNK) cells and liver-resident NK (LrNK) cells, and the latter is also known as liver type 1 ILCs (ILC1s)[4]. Phenotypically, mouse LrNK cells can be identified as NK1.1+NKp46+CD49a+CD49b−, distinct from NK1.1+NKp46+CD49a−CD49b+ cNK cells[4]. Likewise, a CD56^bright^CD16^low^CD49a+ LrNK cell subset was identified in human liver[5]. LrNK cells reside in hepatic sinusoid and exhibit significant differences in terms of development, phenotype and effector functions compared to cNK cells[6]. Firstly, although cNK cells develop from the progenitors in the bone marrow, LrNK cells are generated from liver Lin−CD122+CD49a+ progenitors via an interferon (IFN)-γ-dependent loop[7]. Secondly, the development and maintain of LrNK cells uniquely rely on a panel of specific transcription

[1]Key Laboratory for Experimental Teratology of Ministry of Education, Key Laboratory of Infection and Immunity of Shandong Province and Department of Immunology, School of Basic Medical Sciences, Cheeloo Medical College of Shandong University, Jinan 250012 Shandong, China. [2]Collaborative Innovation Center of Technology and Equipment for Biological Diagnosis and Therapy in Universities of Shandong, Jinan 250012 Shandong, China. [3]Key Laboratory for Experimental Teratology of Ministry of Education, Department of Histology and Embryology, School of Basic Medical Science, Shandong University, Jinan, China. ✉e-mail: machunhong@sdu.edu.cn; liangxiaohong@sdu.edu.cn

factors, including T-bet, PLZF, Hobit, AhR and RORα[8–14]. Functionally, LrNK cells produce higher level of tumor necrosis factor (TNF)-α, lower level of IFN-γ and perforin, and have similar levels of granzyme B as compared to cNK cells[14]. Moreover, LrNK cells not only participate in anti-tumor immunity[14], but also mediate local immune tolerance[15] and immune memory[16,17]. Thus, further unraveling the mechanisms by which the liver microenvironment supports the development and functional specialization of LrNK cells would provide insights for understanding the liver biology and developing therapeutic strategies for liver diseases.

The liver is at the nexus of host-microbial interactions with respect to its unique anatomical location, allowing continuous blood flow from the gastrointestinal tract through the liver sinusoids[18]. Importantly, accumulating evidence unraveled that gut microbiota plays crucial roles in the establishment and maintenance of liver immune homeostasis. Gut commensal and microbial products such as lipopolysaccharide (LPS) induce the sustained MYD88-dependent signaling, which in turn orchestrate the polarized distribution of Kupffer cells and NKT cells concentrating around periportal regions and optimizes the effective host defense[19]. On the other hand, commensal bacteria are critical for maintaining Kupffer cells in a tolerant state, preventing subsequent NKT cell over-activation during liver regeneration[20]. Moreover, commensal lipid antigens presented by hepatocytes is required for sustaining liver-resident γδT-17 cell homeostasis, including activation, survival and proliferation[21]. However, although germ-free (GF) mice showed either reduced or enhanced activity of circulating NK cells in different infection models compared to specific pathogen free (SPF) mice[22,23], the importance of gut commensal in the regulation of hepatic NK cells remains unexplored.

Early life is a critical period for establishing a healthy gut microbiota which imprints the immune system and persists long even into adulthood[24–26]. However, by one year, up to 50% of infants will have suffered from the microbiota dysbiosis, especially due to antibiotic exposure[27]. Strikingly, intestinal dysbiosis in infant mice impairs antibody responses to vaccines[28]. On the contrary, early life flora disorder causes the hyperactivation of intestinal macrophages and the enhanced inflammatory Th1 responses to bacteria stimulation, leading to the increased risk for inflammatory bowel disease (IBD)[29]. Likewise, mice treated neonatally with antibiotics develop exacerbated experimental psoriasis through increased IL-22-producing γδ T cells[30].

Considering the special developmental pathway of LrNK cells in liver and the close interplay between gut and liver, we hypothesized that gut microbiota would have a profound impact on the homeostasis of LrNK cells. By using a maternal antibiotic treatment mouse model, we demonstrated that early-life microbiota depletion persistently blunted functional maturation of LrNK cells even at adulthood. Mechanistically, durative microbiota dysbiosis caused by early-life antibiotic treatment significantly downregulated hepatic butyrate level and IL-18 expression from Kupffer cells and hepatocytes, which in turn impaired functional maturation of LrNK cells. Strikingly, dietary supplementation of experimentally or clinically used *Clostridium butyricum* restored the impaired LrNK cell maturation and function induced by antibiotic exposure at early life. Our study uncovers an interplay network of gut-liver axis for LrNK maturation, highlighting the importance of the early-life microbiota in the development of tissue-resident immune cells.

## Results

### Early-life gut microbiota is crucial for functional maturation of LrNK cells

To determine the impact of gut microbiota on liver NK cell development, we used a maternal antibiotic treatment mouse model[28,31,32]. Briefly, dams and their pups were treated with combination of antibiotics (Abx, including ampicillin, vancomycin, neomycin and metronidazole), in their drinking water in late pregnancy and throughout the pre-weaning mouse infant period, and then fed with normal chow and drinking water (early-Abx) (Fig. 1a). Similar to the literature[33], early-Abx treatment slightly reduced the mouse body weight (Supplementary Fig. 1a). We then characterized LrNK and cNK cells from control or early-Abx mice at weaning or at adult age (8-week-old). Gating strategy and the phenotype characterization of LrNK and cNK cells were shown in Supplementary Fig. 1b, c. There were no significant differences in the percentage and number of both cNK and LrNK cells in early-Abx and control mice (Fig. 1b, Supplementary Fig. 1d). Consistently, there were also no differences in the proliferation and apoptosis of LrNK cells (Supplementary Fig. 1e, f). Intriguingly, compared to those in control mice, LrNK cells in weaning or adult early-Abx mice had decreased mean fluorescence intensity (MFI) and percentage of CD11b and increased CD27 (Fig. 1c, d, Supplementary Fig. 2a), which was demonstrated to be a more immature phenotype[34]. Consistently, LrNK cells of early-Abx mice expressed relatively lower levels of kill cell lectin-like receptor subfamily G member 1 (KLRG1), a molecule associated with NK cell terminal maturation (Fig. 1c, d, Supplementary Fig. 2a). In addition, the expression of transcription factors related with LrNK cell development and maturation, *Rorα* and *Zfp683*, was also downregulated in LrNK cells of early-Abx mice (Supplementary Fig. 2b). In contrast to LrNK cells, cNK cells in early-Abx mice displayed comparable levels of CD11b, CD27 and KLRG1 with control mice (Supplementary Fig. 2c, d). Since LrNK cells, distinct from cNK cells, develop locally from Lin⁻Sca-1⁺Mac-1⁺ hematopoietic stem cells (LSM), which then differentiate into Lin⁻CD122⁺CD49a⁺ cells, as precursor cells for LrNK[7], we thus detected the LSM cells and progenitor cells of liver and bone marrow. Results showed that there were no differences in the percentage and number in LSM cells and progenitor cells between control and early-Abx mice (Supplementary Fig. 2e–g).

Next, we investigated whether the function of LrNK and cNK cells was affected by early-life antibiotics treatment. Upon stimulation with phorbol 12-myristate 13 acetate (PMA) and ionomycin (Ion), LrNK cells of weaning or adult early-Abx mice displayed markedly lower production of effector molecules than those of control mice (Fig. 1e, f, Supplementary Fig. 3a). Reduced expression of IFN-γ were also observed in LrNK cells of early-Abx mice under stimulation of either IL-12/IL-15 or polyriboinosinic: polyribocytidylic acid (poly (I: C)) (Fig. 1g). By using an in vitro flow-based killing assay, LrNK cells of 8-week-old early-Abx mice exhibited significant lower killing activities against Yac-1 cells than those from control mice (Fig. 1h). Moreover, early-life antibiotic exposure led to increased expression of inhibitory receptors Tim-3, PD1, and NKG2A in LrNK cells of weaning or adult mice (Fig. 1i and Supplementary Fig. 3b). Distinct from LrNK cells, there was no significant changes in the expression of effector molecules and inhibitory receptors, as well as cytotoxic activities, of liver cNK cells between adult control and early-Abx mice (Supplementary Fig. 3c–e). In addition, to further disentangle the effect of antibiotics exposure during pregnancy or breastfeeding period on LrNK functional maturation, we introduced utero-Abx or breastfeeding-Abx mouse model. Results showed that antibiotic treatment during pregnancy or breastfeeding period also impaired the maturation and function of LrNK cells (Supplementary Fig. 4a–f). These results strongly suggest that early-life microbiota sustains the homeostasis of LrNK cells.

It is well established in previous studies that blockade of NK cell maturation impairs their anti-tumor potential[35,36]. Most recently, the homeostasis of LrNK cells were demonstrated to be critical for suppressing liver tumor progression[14]. Thus, we wonder whether early-life antibiotics exposure also blunts LrNK-mediated anti-tumor effects. To address this, we introduced both c-myc/AKT-driven[37,38] (Fig. 2a) and STZ-HFD-induced[39] (Supplementary Fig. 5a) hepatocellular carcinoma (HCC) mouse model. It was found that early-Abx

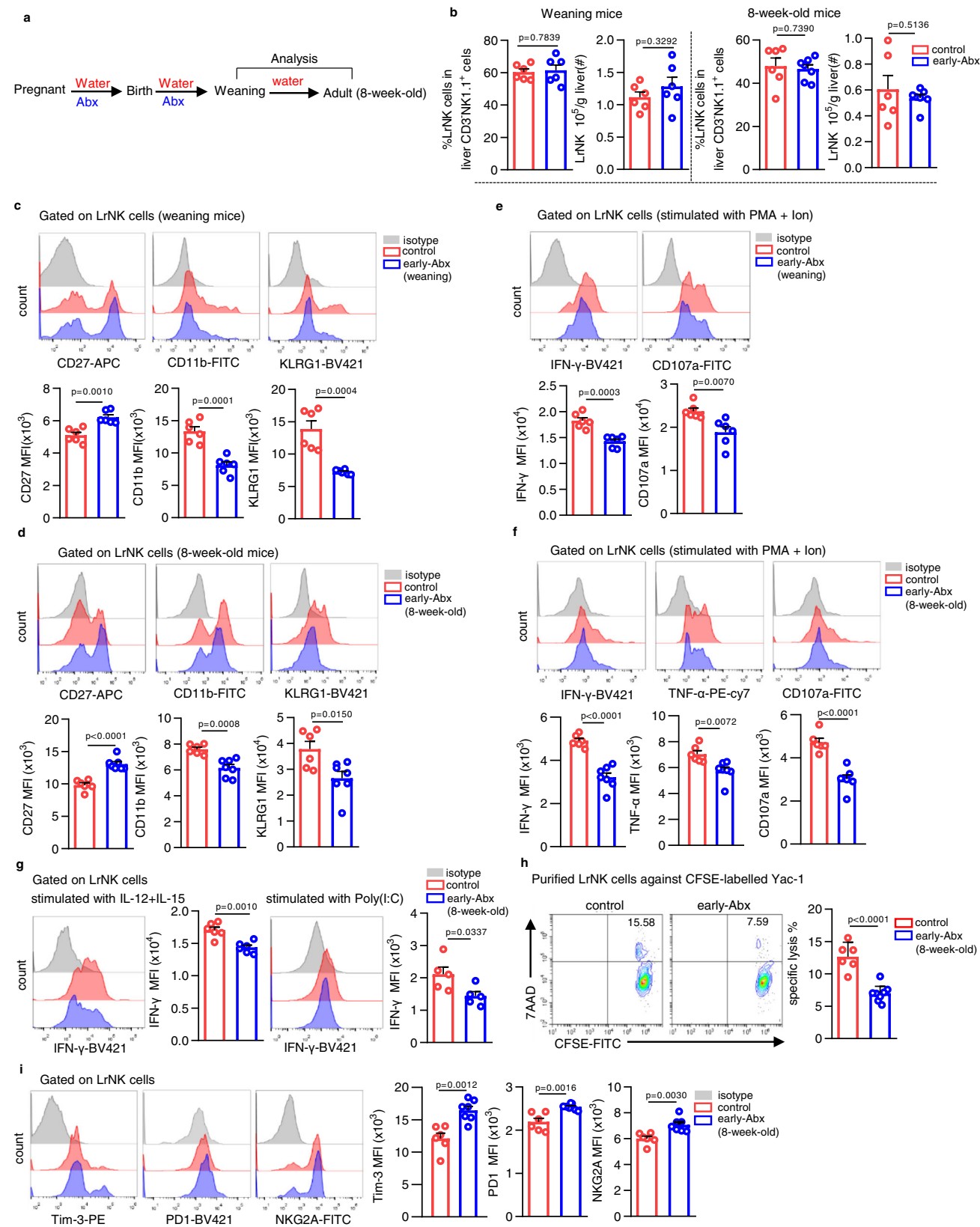

mice developed more severe liver tumors than control mice in both HCC models (Fig. 2b, c, Supplementary Fig. 5b, c). Accordingly, compared to those of control mice, tumor-infiltrating LrNK cells of early-Abx mice had reduced expression of effector molecules, decreased cytotoxic activities, but increased level of inhibitory receptors (Fig. 2d–f, Supplementary Fig. 5d). Importantly, depletion

of both LrNK and cNK cells with anti-NK1.1, but not cNK depletion alone with anti-asialo GM1, almost completely abrogated the shortening of lifespan of HCC-bearing Abx mice. (Fig. 2g, h). Together, these findings suggest that the impaired functional maturation of LrNK cells induced by early-life microbiota depletion contributes to the accelerating HCC development.

**Fig. 1 | Early-life gut microbiota sustains the maturation and function of LrNK cells. a** Schemes of early-Abx mouse model experimental design. **b** Representative FACS plots and bar graphs for the percentages and absolute number of LrNK cells from control and early-Abx mice (weaning mice: $n = 6$ per group; 8-week-old mice: control $n = 6$, early-Abx $n = 7$). **c, d** Representative FACS plots and bar graph for the expression level (MFI) of CD11b, CD27 and KLRG1 in LrNK cell subsets from weaning or adult control and early-Abx mice (weaning mice: control $n = 6$, early-Abx $n = 6$; 8-week-old mice: control $n = 6$, early-Abx $n = 7$). **e, f** Representative FACS plots bar graph and for the expression (MFI) of effector molecules in PMA and Ion-stimulated LrNK cells from weaning or adult control and early-Abx mice (weaning mice: $n = 6$ per group; 8-week-old mice: control $n = 6$, early-Abx $n = 7$). **g** Representative FACS

plots and bar graph for the expression (MFI) of IFN-γ expression in IL-12/15 or poly(I:C) - stimulated LrNK cells from 8-week-old control and early-Abx mice ($n = 6$ per group). **h** Cytotoxicity of LrNK cells against CFSE-labeled YAC-1 cells. 7-AAD⁺ CFSE⁺ cells represented the killed cells (control $n = 6$, early-Abx $n = 8$).
**i** Representative FACS plots and bar graph for the expression (MFI) of Tim-3, NKG2A and PD-1 in LrNK cells from control and early-Abx mice (control $n = 6$, early-Abx $n = 7$). Each symbol represents data from an individual mouse, and error bars represent SEM per group in one experiment. Data were analyzed using two-tailed Student's $t$-test. $^*P < 0.05$; $^{**}P < 0.01$; $^{***}P < 0.001$; $^{****}P < 0.0001$; ns, no significance. All experiments were repeated two or three independent experiments. Source data are provided as a Source Data file.

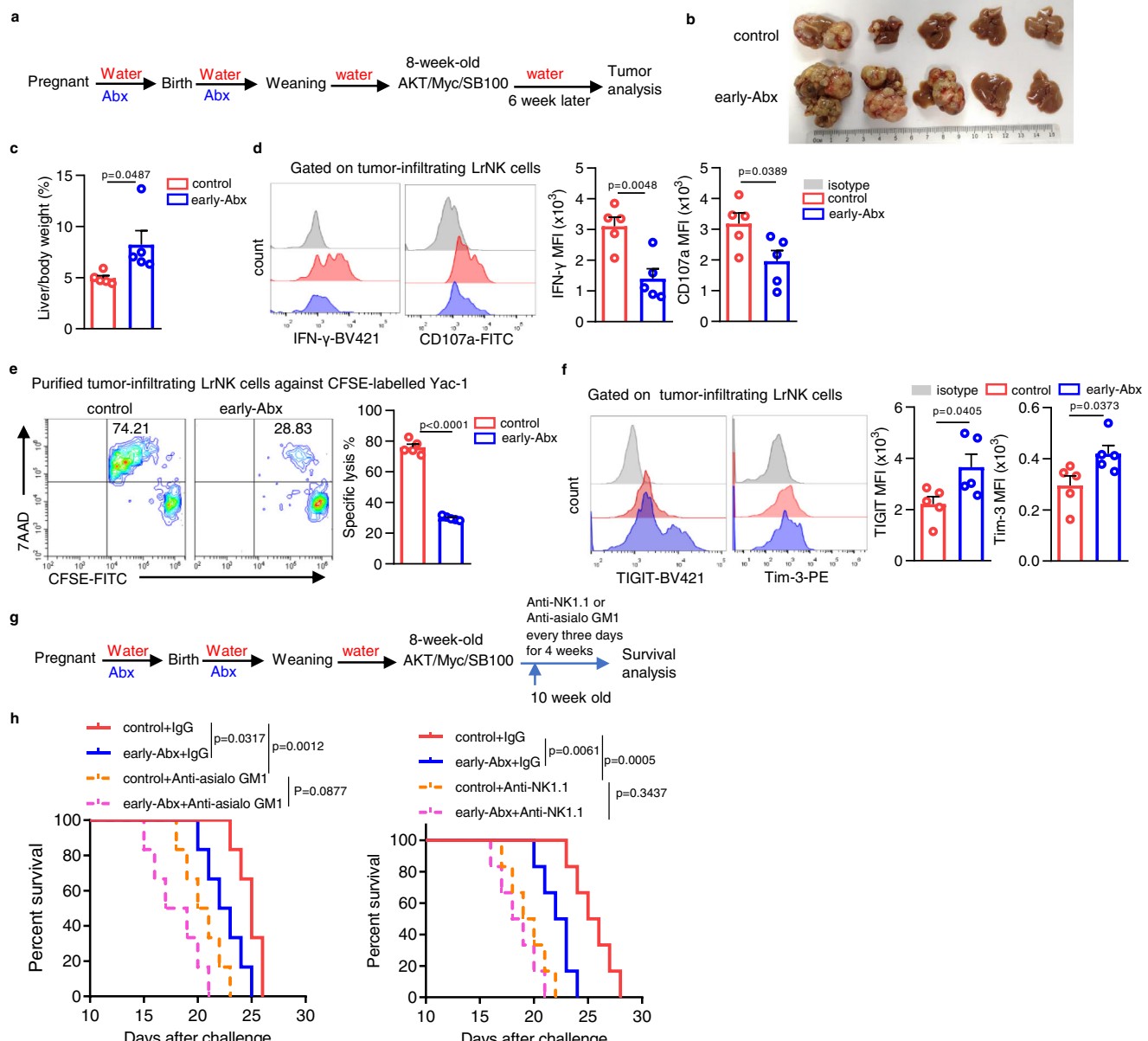

**Fig. 2 | Impaired LrNK function contributes to early-Abx enhanced HCC progression. a** Experimental design of c-myc/AKT/SB100-induced HCC model in control or early-Abx mice. **b, c** In vitro tumor imaging and liver /body weight was shown ($n = 5$ per group). **d** Representative FACS plots and bar graph for the expression (MFI) of IFN-γ and CD107a in LrNK cells from control and early-Abx HCC mice ($n = 5$ per group). **e** Representative FACS plots and bar graph for the cytotoxicity of LrNK cells from control or early-Abx HCC mice against CFSE-labeled YAC-1 cells ($n = 5$ per group). **f** Representative FACS plots and bar graph for the

expression (MFI) of TIGIT and Tim3 on LrNK cells from control or early- Abx HCC mice ($n = 5$ per group). **g, h** Experiment scheme and survival curves of control and early-Abx mice injected with AKT/Myc/SB100 plasmids and treated with anti-asialo GM1 or anti-NK1.1($n = 5$ per group). Dots represent data from individual mice, and error bars represent SEM per group in one experiment. Statistical significance was tested by two-tailed Student's $t$ test (**c**–**f**) or Long-Rank test (**h**). $^*P < 0.05$; $^{**}P < 0.01$; $^{***}P < 0.001$; $^{****}P < 0.0001$; ns, no significance. Source data are provided as a Source Data file.

## The persistent alteration of gut microbiota is responsible for LrNK cell maturation arrest in early-Abx treated mice

It has been demonstrated that the colonization of microbiota in early life has long-term impact on the diversity and stability of intestinal microbial communities. Thus, we wonder whether the persistent gut dysbiosis serves impaired LrNK cell maturation in mice with early-life antibiotics exposure. We analyzed gut microbiota using 16 S rRNA sequencing with feces from 8-week-old control or early-Abx mice. Feces of early-Abx mice contained significantly less α-diversity than the control group, indicated as decreased index of operational taxonomic unit (OTU) level (Fig. 3a). Microbial culture on blood agar plate and qPCR analysis for fecal bacterial DNA content and total 16 S rRNA also confirmed that early life antibiotic treatment had persistent alteration on the total abundance of gut microbiota (Supplementary Fig. 6a–c). Principal coordinate analysis (PCoA) plots also revealed significant discrete clustering in microbial community structure (β-diversity) of control and early-Abx groups (Fig. 3b). Further, linear discriminant analysis effect size (LEfSe) showed that the mean relative abundance of top 15 abundant bacteria was significantly different between control and early-Abx mice (Fig. 3c, d). Especially, some probiotic bacteria such as *Akkermansia*, *Bifidobacterium*, *Prevotellaceae*, *Allobaculum* were significantly reduced in early-Abx mice (Fig. 3c, d). These verify that early-life antibiotic treatment significantly alters microbial community composition in adulthood. Next, we co-housed early-Abx mice for 5 weeks with control littermates immediately after weaning (Fig. 3e), as previously reported[30]. As expected, co-housing treatment eliminated the differences in the expression of CD27, CD11b and KLRG1 in LrNK cells between control and early-Abx mice (Fig. 3f, Supplementary Fig. 6d). Consistently, there were comparable levels of IFN-γ production and CD107a mobilization of LrNK cells between cohousing control and early-Abx mice (Fig. 3g, Supplementary Fig. 6e). These findings suggested that the disruption of commensal microbiota by early-life antibiotic treatment accounts for the impaired functional maturation of LrNK cells.

## Microbiota-derived butyrate facilitates functional maturation of LrNK cells

The gut microbiota can impact the physiology of the host by generating abundant commensal metabolites[40]. We, therefore, turned to evaluate the roles of microbiota-derived metabolites in the impairment of LrNK maturation in early-Abx mice. Liquid chromatograph-mass spectrometer (LC-MS) revealed the difference in abundance and clustering of fecal metabolites between 8-week-old control and early-Abx mice (Supplementary Fig. 7a, b). KEGG pathway showed the enrichment of differentially expressed metabolites in multiple metabolic pathways, including lipid metabolism, amino acid metabolism and carbohydrate metabolism (Supplementary Fig. 7c). Considering the importance of short-chain fatty acids (SCFAs) in regulating immune homeostasis, we further detected their levels in feces of control and early-Abx mice by Gas Chromatography-Mass Spectrometer (GC-MS). Results showed that several SCFAs, such as butyrate, were significantly reduced in feces of early-Abx mice (Fig. 4a). Notably, butyrate, but not acetate and propionate, had close correlation with the abundance of the dominant bacterial in early-Abx mice (Fig. 4b). Moreover, compared to that of control mice, feces of early-Abx mice had significantly decreased abundance of butyrate-producing microbiome like *Faecalibacterium*, *Roseburia*, *Fusobacteria*, and *Eubacterium*[41] (Fig. 4c). These indicate that early-Abx treatment reduces the production of butyrate in adult mice.

Next, we wonder whether microbiota-derived butyrate is responsible for the maturation of LrNK cells. Interestingly, the content of butyrate in liver of Abx mice was significantly lower than that of control mice, while there was comparable level of butyrate in spleen of two groups of mice (Fig. 4d). Moreover, administration of butyrate (Fig. 4e) partially or completely recovered the expression of CD27,

CD11b and KLRG1 in LrNK cells of early-Abx mice to the level of control mice (Fig. 4f, Supplementary Fig. 7d). In accordance, butyrate supplementation improved IFN-γ expression and CD107a mobilization of LrNK cells from early-Abx mice (Fig. 4g, Supplementary Fig. 7e). In addition, we analyzed the correlation of the maturation and function of LrNK cells with butyrate level in normal liver tissues from hepatic hemangioma patients. We found that, consistent with the mouse experimental data, LrNK cells from butyrate-high liver tissues had lower expression of CD27, but higher level of IFN-γ expression and CD107a mobilization than those cells from butyrate-low livers (Fig. 4h–j). Together, these results support the hypothesis that microbiota metabolite butyrate promotes the maturation of LrNK cells.

## Butyrate indirectly enhances LrNK cell functional maturation through acting on Kupffer cells and hepatocytes by GPR109A

To further determine whether microbiota-derived butyrate influences LrNK cell functional maturation in a cell-intrinsic or extrinsic manner, we transferred purified NK cells from adult control or early-Abx mice (CD45.2) into recipient CD45.1 mice and measured the activity of donor-derived LrNK cells after application of poly(I:C) (Fig. 5a). We found that transferred LrNK cells from control and early-Abx mice had comparable IFN-γ expression and CD107a mobilization (Fig. 5b). In contrast, when we transferred NK cells (CD45.2) separately into control or early-Abx mice (CD45.1) (Fig. 5c), donor LrNK cells in early-Abx mice had lower level of IFN-γ and CD107a than those in control mice (Fig. 5d). Consistently, ex vivo exposure to butyrate did not alter the IFN-γ and CD107a level of LrNK cells (Fig. 5e). These data suggest that early-life gut microbiota affects the functional maturation of LrNK cells in a cell-extrinsic manner.

Multiple cell components including hepatocytes and liver mononuclear cells (LMNCs) consists of the complicated liver microenvironment. To test their potential involvement in LrNK cell functional maturation regulated by microbiota-butyrate axis, we purified LrNK cells and co-cultured with hepatocytes or LMNCs with/without the presence of butyrate respectively. We found that butyrate treatment indeed increased IFN-γ production of LrNK cells when co-cultured with both hepatocytes and LMNCs (Fig. 5f). Butyrate could act either by inhibiting histone deacetylases (HDACs), or through G protein-coupled receptors (GPCRs), such as GPR41, GPR43, and GPR109a[42–44]. However, HDAC inhibitor, Trichostatin A (TSA), did not affect the upregulated expression of IFN-γ in co-cultured LrNK cells by butyrate treatment (Fig. 5f, Supplementary Fig. 8a). Then, we wonder whether GPCRs take effects in this process. RT-qPCR showed that although GPR41, GPR43 and GPR109A were all expressed in LMNCs, only GPR109A was detected in hepatocytes (Fig. 5g). Further, GPR109A knockdown in hepatocytes dampened the augmented IFN-γ and CD107a expression in co-cultured LrNK cells induced by butyrate treatment (Fig. 5h, Supplementary Fig. 8b). For LrNK cells co-cultured with LMNCs, the GPR109A agonist MK-0354, but not the agonists for GPR41 and GPR43 (AR-420626 and 4-CMTB), increased their expression of IFN-γ and CD107a (Supplementary Fig. 8c). Notably, among different subsets of LMNCs, GPR109A was highly expressed in Kupffer cells (Fig. 5i). Moreover, co-culture of LrNK cells with Kupffer cells upregulated the expression of effector molecules of LrNK cells with the treatment of butyrate (Fig. 5j, Supplementary Fig. 8d), while GPR109A silencing abrogated this promoting effect (Fig. 5k, Supplementary Fig. 8e). Together, these findings support the view that microbiota-butyrate axis indirectly promotes LrNK function by acting on GPR109A in hepatocytes and Kupffer cells.

## Butyrate triggers IL-18 production in hepatocytes / Kupffer cells to improve functional maturation of LrNK cells

To gain insight into the mechanism underlying the indirect action of microbiota-butyrate on LrNK cell functional maturation, we examined

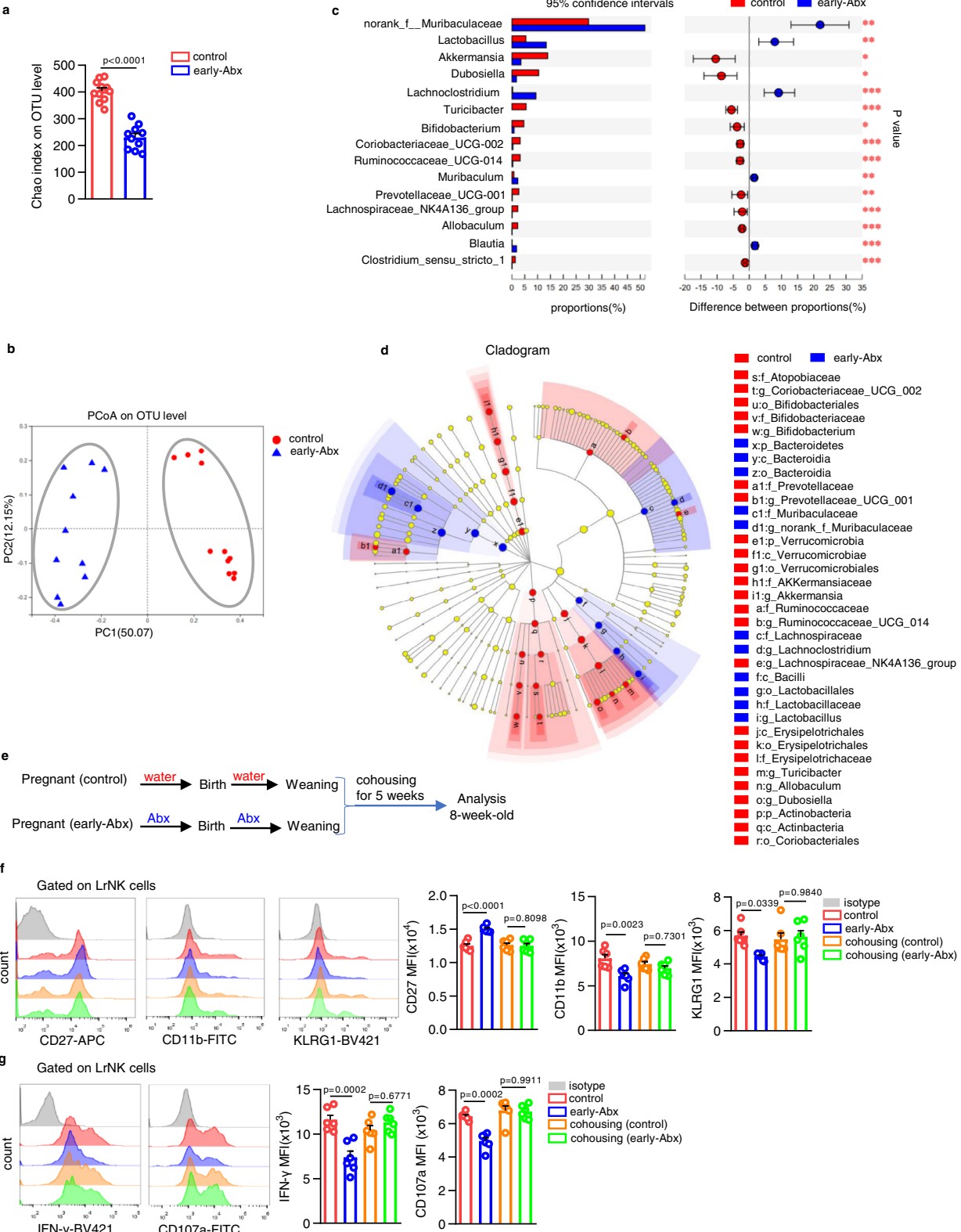

**Fig. 3 | The persistent microbiota dysbiosis is responsible for maturation arrest of LrNK cells in early-Abx-treated mice. a**–**d** 16 S rRNA sequencing was performed with feces from control and early-Abx mice (control $n = 11$, early-Abx $n = 10$). Alpha diversity comparison via chao index analysis and beta diversity comparison via PCoA analysis. **c** Genus level comparison of gut commensal microflora between control (red) and early-Abx (blue) mice. **d** LEfSe (linear discriminant analysis size effect) predictions for bacterial families in fecal pellets of control (red) and early-Abx (blue) mice were shown. **e** Experimental scheme of cohousing mouse model. **f** Representative FACS plots and bar graph for the

expression (MFI) of CD27, CD11b, and KLRG1 on LrNK cells from the four groups of mice ($n = 6$ per group). **g** Representative FACS plots and bar graph for the expression (MFI) of IFN-γ and CD107a in PMA and Ion-stimulated LrNK cells from the four groups of mice ($n = 8$ per group). Dots represent data from individual mice, and error bars represent SEM per group in one experiment. Statistical analysis was performed by two-tailed Student's $t$-test (**a**), two-tailed Wilcoxon rank sum test (**c**), non-parametric factorial Kruskal–Wallis (KW) sum-rank test (**d**) or one-way ANOVA with Tukey's multiple comparisons test (**f**, **g**). $^*P < 0.05$; $^{**}P < 0.01$; $^{***}P < 0.001$; $^{****}P < 0.0001$; ns, no significance. Source data are provided as a Source Data file.

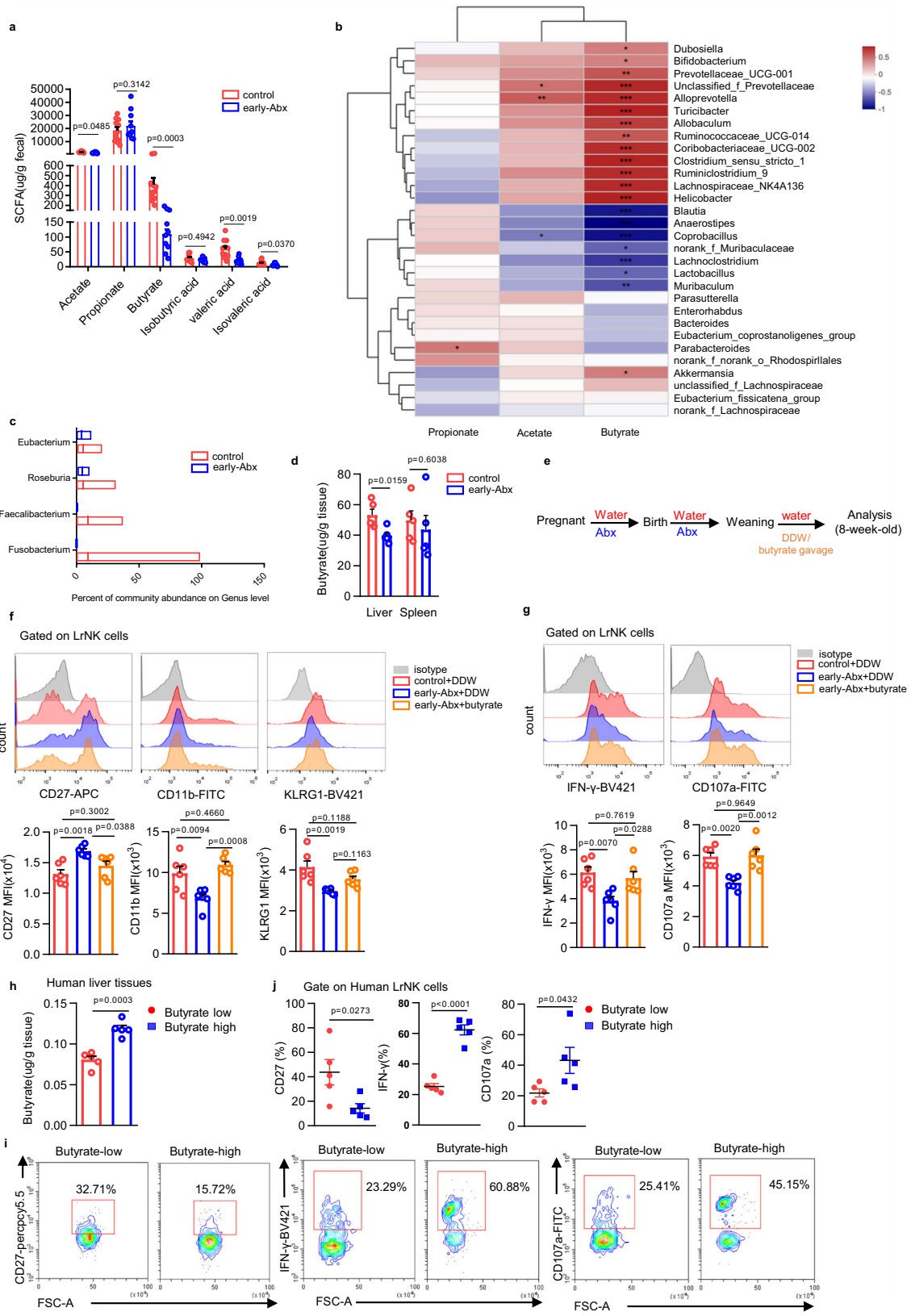

the expression of a panel of cytokines/chemokines in liver tissues from control or early-Abx mice. RT-qPCR analysis showed that the expression of IL-18 was obviously downregulated in early-Abx mice (Fig. 6a). In accordance, IL-18 protein was reduced in both liver and small intestine of early-Abx mice, while the level of IL-18 in liver was significantly higher than that in small intestine (Fig. 6b). Intriguingly,

intraperitoneal injection of recombinant IL-18 protein (rIL-18) markedly rescued the expression of maturation markers, CD11b and KLRG1, as well as the level of effector molecules in LrNK cells from early-Abx mice (Fig. 6c, d, Supplementary Fig. 9a, b). These results suggest the critical role of IL-18 in the gut microbiota-mediated regulation of LrNK maturation.

**Fig. 4 | Microbiota-derived butyrate facilitates functional maturation of LrNK cells. a** GC -MS analysis of short-chain fatty acids (SCFAs) was performed with feces from control or early-Abx mice (control $n = 11$, early-Abx $n = 10$). **b** The correlation heatmap between dominant bacteria and SCFAs in faeces through correlation numerical visualization. **c** The percentage of butyrate-producing bacteria on genus level from 16 S rRNA sequencing of faeces from control and early-Abx mice (control $n = 11$, early-Abx $n = 10$). The boxplot represents the median shown as a line in the center of the box, the boundaries are the first and third quartile, and whiskers represent the minimum and maximum values in the data. **d** Levels of butyrate in liver and spleen tissues from control and early-Abx mice by GC-MS analysis ($n = 5$ per group). **e** Experimental schemes of butyrate gavage mouse model. **f** Representative FACS plots and bar graph for the expression (MFI) CD27, CD11b,

and KLRG1 on LrNK cells from different groups of mice ($n = 6$ per group). **g** Representative FACS plots and bar graph for the expression (MFI) of IFN-γ and CD107a in PMA and Ion-stimulated LrNK cells from different groups of mice ($n = 6$ per group). **h** Grouping according to butyrate level in normal liver tissue from hepatic hemangioma patients. **i, j** Representative FACS plots and bar graph of the percentage of CD27, IFN-γ and CD107a in LrNK cells from patients with high levels of butyrate and those with low butyrate levels. Dots represent data from individual mice or human patients, and error bars represent SEM per group in one experiment. Statistical analysis was performed by two-tailed Student's t-test (**a, d**), parametric factorial Kruskal−Wallis (KW) sum-rank test (**b**) or one-way ANOVA with Tukey's multiple comparisons test (**f, g**). $^*P < 0.05$; $^{**}P < 0.01$; $^{***}P < 0.001$; $^{****}P < 0.0001$; ns, no significance. Source data are provided as a Source Data file.

Then, we wonder whether microbiota-butyrate axis promotes LrNK maturation through upregulating IL-18 in liver microenvironment. Supplementation of butyrate increased IL-18 level in liver (Supplementary Fig. 9c). Consistent with the involvement of hepatocytes and Kupffer cells in butyrate-mediated enhancement of LrNK functional maturation, addition of butyrate upregulated IL-18 level in both hepatocytes and Kupffer cells ex vivo (Supplementary Fig. 9d, e). Importantly, blockade with specific IL-18 antibody (anti-IL-18) almost abolished butyrate-induced upregulation of KLRG1 and effector molecules in LrNK cells co-cultured with hepatocytes or Kupffer cells (Fig. 6e, f). These data suggest that IL-18 is crucial for the microbiota-butyrate axis mediated improvement of LrNK cell maturation.

### IL-18 / IL18R promotes LrNK maturation through improving mitochondrial oxidative phosphorylation

IL-18 has been well-known for its enhancement on NK cell activity[45]. However, little is known about its regulatory action and mechanism in LrNK cell maturation. Profiling of maturation markers disclosed that LrNK cells expressing IL-18 receptor α chain (IL-18Rα⁺ LrNK) had higher expression of CD11b and KLRG1 than IL-18Rα⁻ LrNK cells (Fig. 7a). Accordingly, there were relatively high levels of IFN-γ and CD107a in IL-18Rα⁺ LrNK cells, when compared to IL-18Rα⁻ subset (Fig. 7b). Further, transfer of IL-18Rα-deficient LrNK cells into control or early-Abx mice showed the comparable expression of IFN-γ and CD107a (Fig. 7c, d). These findings further support the hypothesis that IL-18 / IL18R axis accounts for LrNK cell maturation, which is disrupted by early-Abx treatment.

To better understand how IL-18 / IL-18R facilitates the functional maturation of LrNK cells, we performed RNA sequencing for IL-18Rα-positive and -negative liver CD3⁻NK1.1⁺ cells. Analysis of this dataset identified 2420 differentially expressed genes (false discovery rate [FDR] < 0.05; 2-fold change or greater) (Supplementary Fig. 10a) and gene-set enrichment analysis (GSEA) revealed that transcriptional signatures associated with mitochondrial oxidative phosphorylation pathway was positively enriched in IL-18Rα⁺ CD3⁻NK1.1⁺ cells (Supplementary Fig. 10b). We, therefore, measured mitochondrial oxidative phosphorylation in IL-15 activated IL-18Rα⁺ and IL-18Rα⁻ LrNK cells before and after the addition of Oligomycin (Oligo), fluoro-carbonyl cyanide phenylhydrazone [FCCP], rotenone [Rot] and antimycin A [AA]) (Fig. 7e). We detected a relatively high level of basal and maximal oxygen consumption rate (OCR) as well as ATP production in IL-18Rα⁺ LrNK cells (Fig. 7e). Further, flow cytometry analysis showed that IL-18Rα⁺ LrNK cells had higher mitochondrial membrane potential (MMP), indicated by Mitotracker Red staining (Fig. 7f), while lower mitochondrial ROS levels as measured by MitoSOX, than IL-18Rα⁻ LrNK cells (Fig. 7g). Conversely, LrNK cells from early-Abx mice showed decreased MMP and increased mitochondrial ROS, when compared to those cells from control mice (Supplementary Fig. 10c, d). Intriguingly, inhibition of mitochondrial oxidative phosphorylation activity by Rotenone significantly decreased IFN-γ production and abolished the difference between IL-18Rα⁺ and IL-18Rα⁻ LrNK cells (Fig. 7h, Supplementary Fig. 10e). Similar results were observed in LrNK cells from

control and early-Abx mice (Supplementary Fig. 10f). Together, these findings indicate that IL-18/IL-18R enhances LrNK functional maturation through improving mitochondrial activity.

### Dietary *Clostridium butyricum* rescues the impaired maturation of LrNK cells in early-Abx-treated mice

Then we resorted to demonstrate whether dietary supplementation of butyrate-producing bacteria could recover the status of LrNK cells from early-Abx treated mice. *Clostridium butyricum* (*C. butyricum*), a butyrate - producing human gut symbiont that has been safely used as a probiotic for decades[46], was administrated to early-Abx mice after weaning (Fig. 8a). Strikingly, *C. butyricum* treatment significantly decreased the expression of CD27, but increased CD11b and KLRG1 level, of LrNK cells from early-Abx mice (Fig. 8b, Supplementary Fig. 11a). In accordance, supplementation of *C. butyricum* enhanced IFN-γ production and CD107a mobilization of LrNK cells in early-Abx mice (Fig. 8c, Supplementary Fig. 11b). Moreover, hepatic IL-18 expression in *C. butyricum* treated early-Abx mice was almost recovered to the comparable level of control mice (Fig. 8d). Intriguingly, dietary supplementation of *Clostridium butyricum* Powder ($2 \times 10^6$/ 0.2 ml per day) (trade name: Baolean, Qingdao Donghai Pharmaceutical Co., Ltd), a commercially probiotic preparation which has been commonly used to treat infantile diarrhea in clinics, partially or completely recovered the expression of CD27, CD11b and KLRG1, as well as IFN-γ level, in LrNK cells of early-Abx mice (Fig. 8e, f, Supplementary Fig. 11c, d). Together, these results preliminarily suggest that dietary supplementation of *C. butyricum* is a potential intervention strategy for rescuing LrNK cell maturation after early exposure to Abx.

## Discussion

Gut microbial development in the first year of life occurs concomitantly to the establishment of our immune system and forms an interactive signaling network[47]. LrNK cells reside in liver sinusoids and are supposed to be potentially regulated by commensal bacterial or their metabolites. In this study, we defined that early-life gut microbiota sustains the functional maturation of LrNK cells. Mechanistically, this regulatory effect is achieved through butyrate production and the subsequent IL-18 production by hepatocytes and Kupffer cells. Our data reveal a mechanism by which gut microbiota influence hepatic immune homeostasis and underlines the long-lasting effects ensuing antibiotics administration at early life.

LrNK cells develop from liver Lin⁻CD122⁺CD49a⁺ progenitors[7] and then mature into CD27⁻ population through CD27⁺ counterparts[34]. The local development model of LrNK cells clue that liver microenvironment may be critical for this process; however, the regulatory mechanisms remain largely unknown. Gut microbial and their metabolites from portal vein exhibit profound impacts on liver physiology and immune homeostasis. In the present study, we found that early antibiotics treatment hindered the maturation and anti-tumor activity of LrNK cells even at adulthood, while liver cNK and splenic NK cells remained unaffected. Co-housing treatment abrogated the inhibitory effect of early-Abx exposure on LrNK cells, indicating the decisive roles

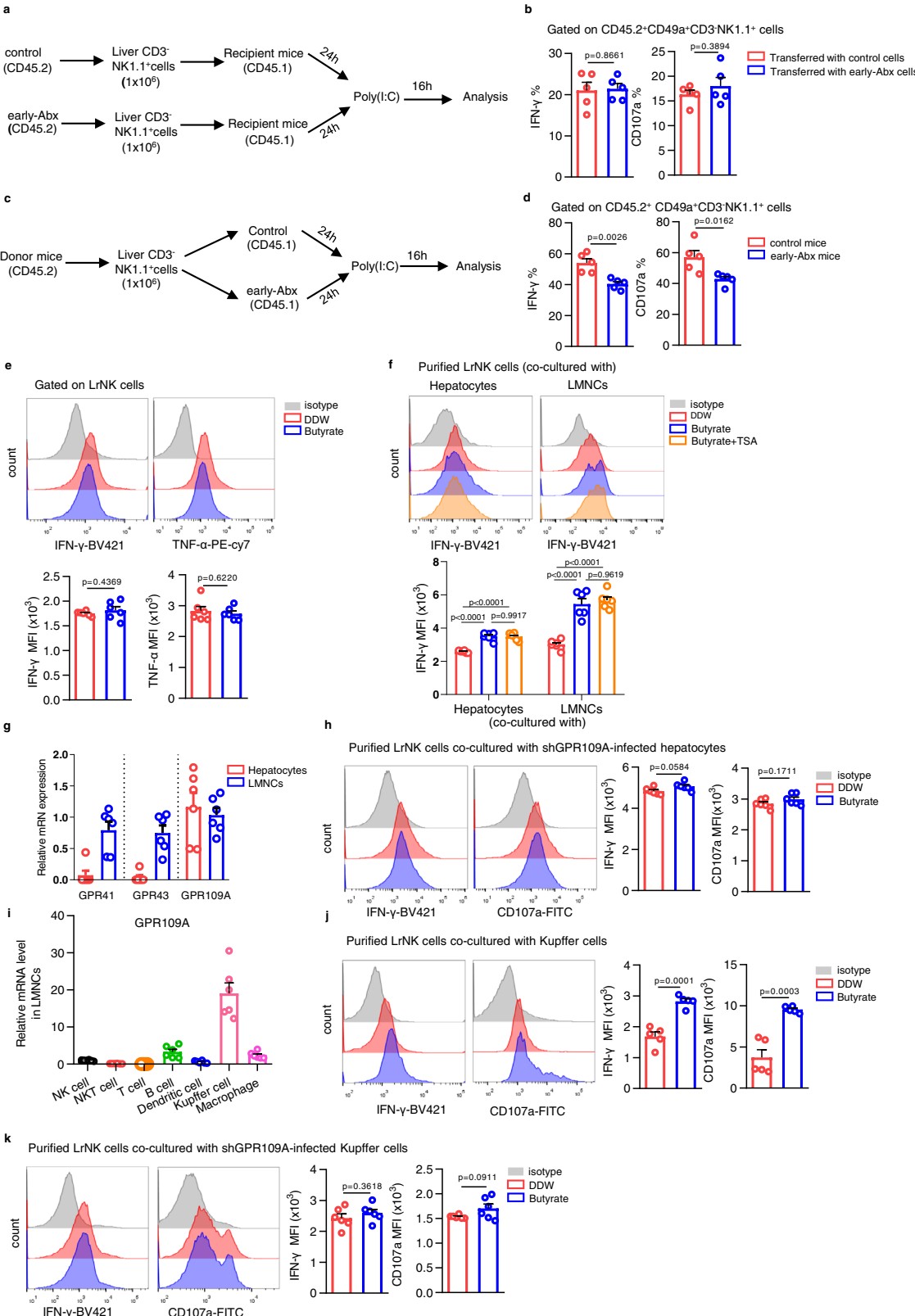

of gut commensal dysbiosis. This persistent impact of early-Abx treatment on LrNK cell maturation further substantiate the importance of early-life microbiota colonization on the development of immune system. Further, adoptive transfer experiments revealed that there was no LrNK-cell intrinsic defect in early-Abx mice. These insights highlight that early-life antibiotic exposure disrupts the liver microenvironment and in turn leads to the persistent maturation arrest of LrNK cells.

However, the disparity in the regulation of LrNK and cNK cells by early-life gut microbiota needs to be further investigated and roles of other factors related with human early-life microbiota perturbations, such as way of birth, feeding, in LrNK cells regulation remains largely unknown. In addition, antibiotics exposure was reported to potentially link with the increased incidence of tumors, including HCC[48], which is in compliance with the involvement of impaired LrNK cell function in HCC

**Fig. 5 | Butyrate indirectly enhances LrNK function through acting on Kupffer cells and hepatocytes through GPR109A. a, b** Experimental scheme and FACS analysis of IFN-γ and CD107a expression in transferred LrNK cells from control or early-Abx mice (CD45.2) after injection of poly(I:C) (*n* = 6 per group).
**c, d** Experimental scheme and FACS analysis of IFN-γ and CD107a expression in LrNK cells (CD45.2) transferred into control or early-Abx mice (CD45.1) after injection of poly(I:C) (*n* = 6 per group). **e** Representative FACS plots and bar graph for the expression (MFI) of IFN-γ and TNF-α in LrNK cells were shown in DDW or butyrate. (*n* = 6 per group). **f** Representative FACS plots and bar graph for the expression (MFI) of IFN-γ in LrNK cells in different groups were shown (*n* = 6 per group). **g** RT-qPCR analysis for GPR41, GPR43 and GPR109A expression level in LMNCs or hepatocytes (*n* = 6 per group). **h** Purified LrNK cells from C57BL/6 mice were co-cultured with hepatocytes infected with GPR109A shRNA lentivirus (Lv-shGPR109A), treated with DDW or butyrate. Representative FACS plots and bar

graph for the expression (MFI) of IFN-γ and CD107a in LrNK cells were shown (*n* = 6 per group). **i** RT-qPCR analysis of GPR109A expression level in different immune cell subsets in liver (*n* = 6 per group). **j** Representative FACS plots and bar graph for the expression (MFI) of IFN-γ and CD107a in LrNK cells were shown co-cultured with Kupffer cells (*n* = 5 per group). **k** Purified LrNK cells from C57BL/6 mice were co-cultured with Kupffer cells infected with GPR109A shRNA lentivirus (Lv-shGPR109A), treated with DDW or butyrate. Representative FACS plots and bar graph for the expression (MFI) of IFN-γ and CD107a in LrNK cells were shown (*n* = 6 per group). Dots represent data from individual mice, and error bars represent SEM per group in one experiment. Data were analyzed using two-tailed Student's *t* test (**b**, **d**, **e**, **h**, **j**, **k**) or one-way ANOVA with Tukey's multiple comparisons test (**f**). *P < 0.05; **P < 0.01; ***P < 0.001; ****P < 0.0001; ns, no significance. Source data are provided as a Source Data file.

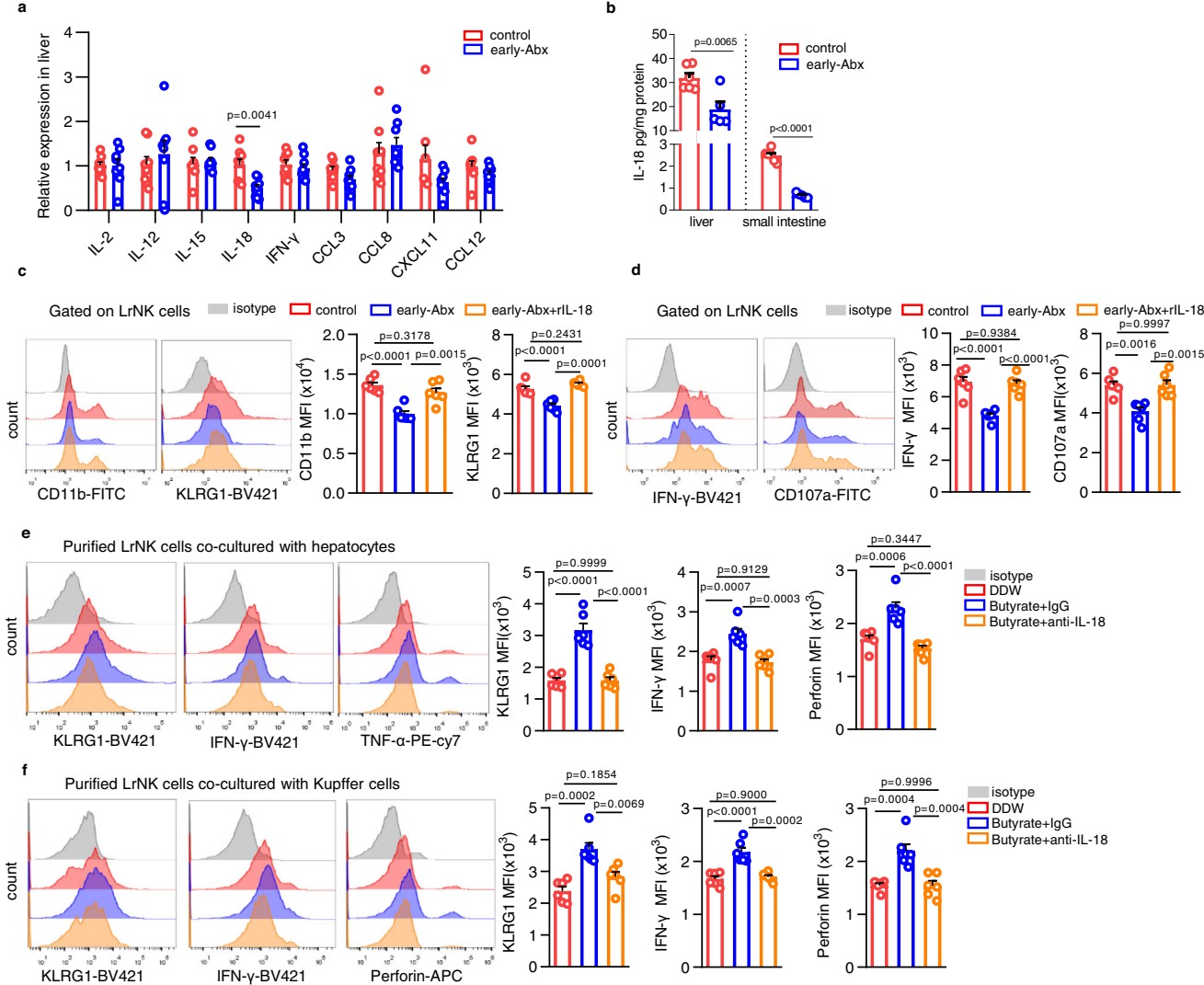

**Fig. 6 | Butyrate triggers IL-18 production in hepatocytes / Kupffer cells to improve functional maturation of LrNK cells. a** RT-qPCR analysis of indicated cytokines/chemokines expression level in liver tissues from control or early-Abx mice (*n* = 8 per group). **b** ELISA assay for IL-18 levels in livers and small intestines of control (*n* = 6 per group) and early-Abx (*n* = 5 per group) mice. **c, d** Representative FACS plots and bar graph for the expression (MFI) of maturation markers or effector molecules in LrNK cells from control or early-Abx mice with or without recombinant IL-18 treatment (*n* = 6 per group). **e, f** Representative FACS plots and

bar graph for the expression (MFI) of KLRG1, IFN-γ and TNF-α or perforin in LrNK cells co-cultured with hepatocytes or Kupffer cells in the presence of butyrate and/ or anti-IL-18 (*n* = 6 per group). Dots represent data from individual mice, and error bars represent SEM per group in one experiment. Statistical analysis was performed by two-tailed Student's *t*-test (**a**, **b**) or one-way ANOVA with Tukey's multiple comparisons test (**c–f**). **P < 0.01; ***P < 0.001; ****P < 0.0001; ns, no significance. Source data are provided as a Source Data file.

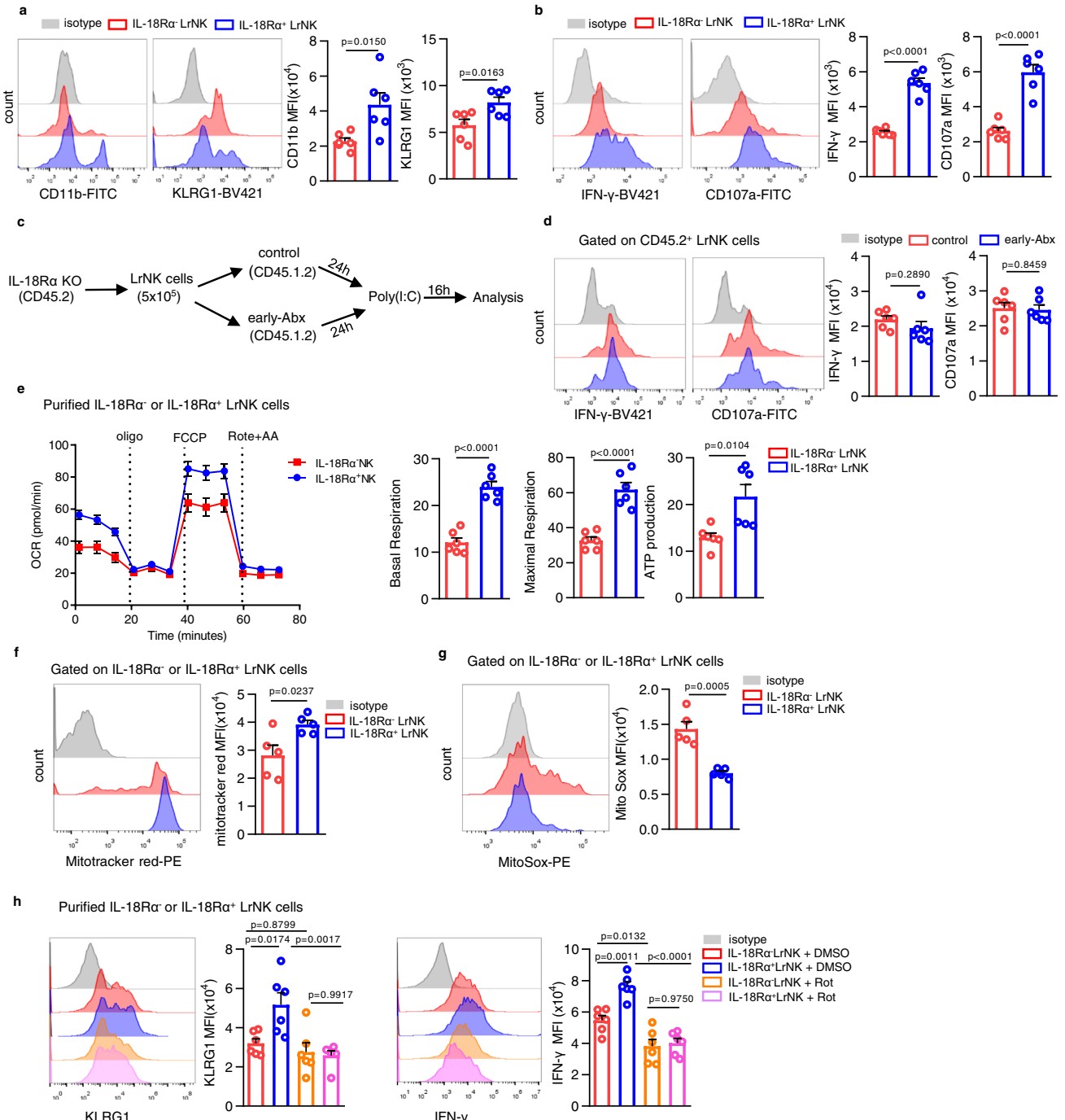

**Fig. 7 | IL-18 / IL18R promotes LrNK maturation through improving mitochondrial oxidative phosphorylation. a, b** Representative FACS plots and bar graph for the expression (MFI) of maturation markers (A) or effector molecules (B) in IL-18Rα⁻ LrNK and IL-18Rα⁺ LrNK in C57BL/6 mice (n = 6 per group).
**c, d** Experimental scheme and representative FACS plots and bar graph for the expression (MFI) of IFN-γ and CD107a in IL-18Rα-deficient LrNK cells (CD45.2) transferred into control or early-Abx mice (CD45.1) mice (n = 6 per group).
**e** Representative images for oxygen consumption rate (OCR) analysis of purified IL-18Rα⁺ and IL-18Rα⁻ LrNK cells stimulated with IL-15 ex vivo and the basal respiration, maximum respiration and ATP production were analyzed (n = 6 per group).

**f, g** Representative FACS plots and bar graph for the mitochondrial membrane potential, indicated by Mitotracker Red staining and mitochondrial ROS levels measured by MitoSOX in IL-18Rα⁻ LrNK and IL-18Rα⁺ LrNK cells (n = 5 per group).
**h** Representative FACS plots and bar graph for the expression (MFI) of KLRG1 and IFN-γ in purified IL-18RαLrNK or IL-18Rα⁺LrNK cells treated with DMSO or Rotenone (n = 6 per group). Dots represent data from individual mice, and error bars represent SEM per group in one experiment. Statistical analysis was performed by two-tailed Student's t-test (**a–g**) or one-way ANOVA with Tukey's multiple comparisons test (**h**). *P < 0.05; **P < 0.01; ***P < 0.001; ****P < 0.0001; ns, no significance. Source data are provided as a Source Data file.

development[14]. It would be also interesting to further investigate whether early-Abx treatment-induced LrNK maturation hindrance correlates with the increased risk of human HCC.

Microbial metabolites are important mediators to modulate host immune responses by microbiota[49]. Of the many bacterial metabolites

in the gut, SCFAs including acetate, propionate, and butyrate, are most abundant and have emerged as critical regulators of immune responses[50]. Here we found that early Abx exposure significantly reduced the level of butyrate in both feces and liver, but not in spleen. Furthermore, supplementation of butyrate or butyrate-producing

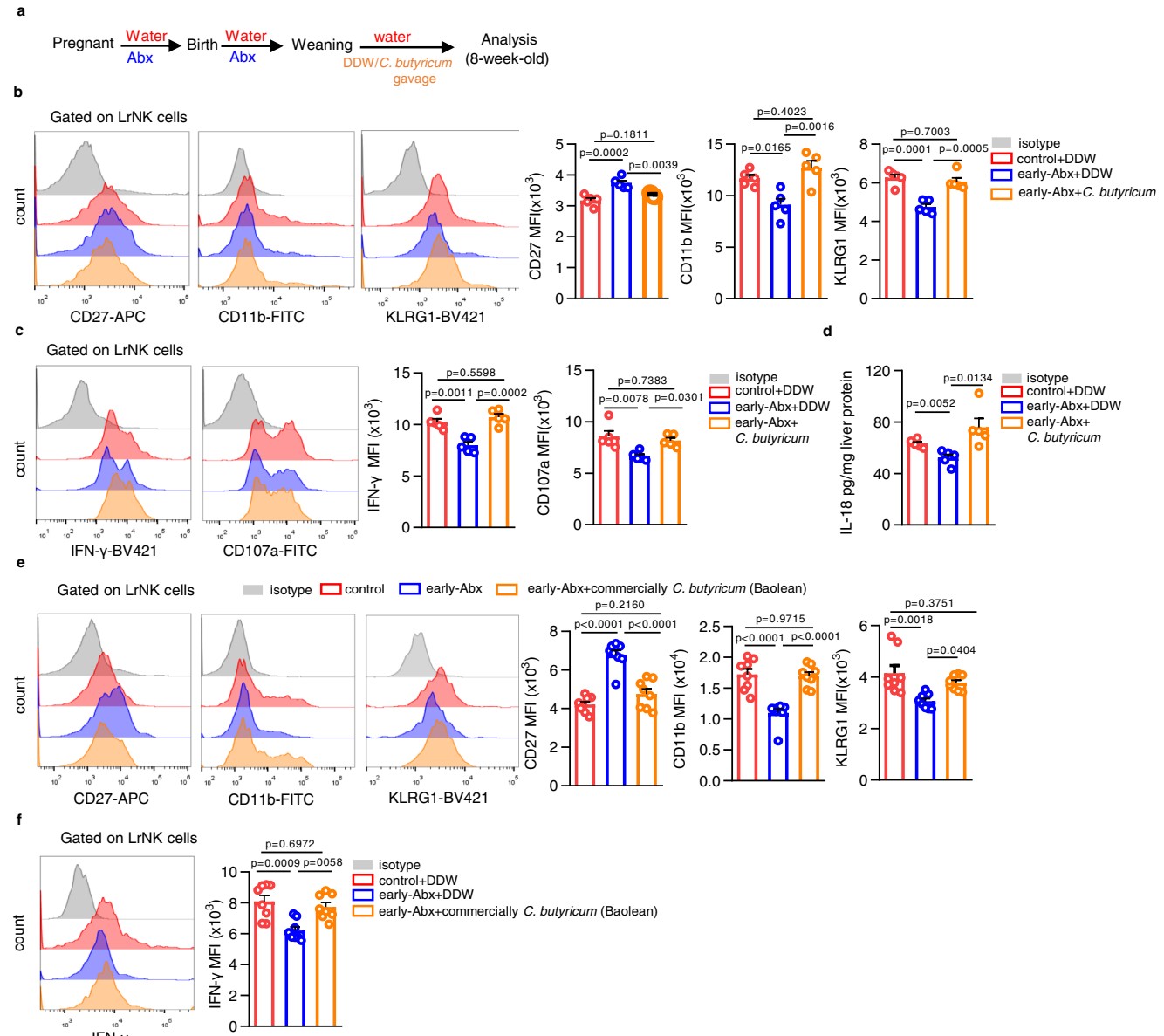

**Fig. 8 | Dietary *Clostridium butyricum* rescues LrNK maturation impairment in early-Abx treated mice. a** Experimental scheme of the *Clostridium butyricum* (C. butyricum) gavage mouse model. **b**, **c** Representative FACS plots and bar graph for the expression (MFI) of maturation markers and effector molecules in LrNK cells from different groups of mice (*n* = 5 per group). **d** ELISA assay of IL-18 levels in liver tissues from different groups of mice (*n* = 5 per group). **e**, **f** Representative FACS plots and bar graph for the expression (MFI) of maturation markers and effector molecules in LrNK cells from different groups of mice (*n* = 8 per group). Dots represent data from individual mice, and error bars represent SEM per group in one experiment. Statistical analysis was performed by one-way ANOVA with Tukey's multiple comparisons test. $^{*}P < 0.05$; $^{**}P < 0.01$; $^{***}P < 0.001$; $^{****}P < 0.0001$; ns, no significance. Source data are provided as a Source Data file.

symbiont, *C. butyricum*, rejuvenated the maturation and function of LrNK cells in early-Abx mice. Recently, butyrate has attracted great attention due to its versatile roles in immune regulation. Provision of butyrate in vivo not only boosted the antitumor CD8[+] T cell response[51], but also promoted the memory potential of antigen-activated CD8[+] T cells[52], and induced the differentiation of macrophages with potent antimicrobial function[53]. Unexpectedly, we did not detect an obvious improvement of butyrate on the function of purified LrNK cells ex vivo. However, the promoting effects of butyrate were recapitulated in LrNK cells co-cultured with hepatocytes or Kupffer cells. It was previously reported that the maturation of LrNK cells was greatly impaired in the absence of CD8[+] T cells, but not CD4[+] T cells or NKT cells[34]. Together with our findings, it is clearly demonstrated that liver microenvironment collaboratively supports the maturation of LrNK cells. Furthermore, here we emphasize the importance of gut microbiota in the

regulatory network for LrNK cell maturation in liver microenvironment. However, apart from butyrate, early-Abx exposure prominently affected a varity of microbial metabolites lipid metabolism, amino acid metabolism and carbohydrate metabolism, all of which has been demonstrated to involve in regulating immune homeostasis[54–56]. The probability that other metabolites might also involves in this inhibition could not be ruled out.

A panel of cytokines have been reported to sequentially regulate the development and maturation of NK cells[2]. IFN-γ produced by LrNK cells was demonstrated to promote their own development from the progenitors[7]. Our transwell co-culture experiment of LrNK cells with hepatocytes or Kupffer cells indicated that the soluble factors involved in the improvement of butyrate-mediated LrNK cell maturation and function. In early-Abx mice, we detected the significant reduction of IL-18 in liver tissues, while butyrate replenishment fixed this deficiency in

a GPR109A-dependent manner. Moreover, recombinant IL-18 protein rescued the maturation and function of LrNK cells in early-Abx mice. IL-18 has been well-known as a stimulator for NK cell activity[45,57]. IL-18R-deficient NK cells were unable to secrete IFN-γ in response to ex vivo stimulation with IL-12, indicating that IL-18 signaling is essential for NK cell priming[58]. In the liver metastatic colon cancer model, IL-18 is required for the suppression of Nlrp3 inflammasome on tumor growth and its promotion of intratumoral NK cell maturation and tumoricidal activity[59]. However, the exact roles of IL-18 in the functional maturation of NK cells remain elusive. Here, we found that compared to IL-18Rα[+] LrNK cells, IL-18Rα[-] LrNK cells exhibited an immature phenotype, accompanied by relatively low mitochondrial activity. Moreover, inhibition of mitochondrial oxidative phosphorylation neutralized the functional difference between IL-18Rα[+] and IL-18Rα[-] LrNK cells. It will be interesting to further interrogate the molecular circuits for IL-18 regulation on the mitochondria activity of LrNK cells. Furthermore, clinical studies showed that IL-18 has the ability to enhance the function of human liver NK cells[60], and human *IL-18* gene polymorphisms were reported to be associated with the risk and severity of HCC[61]. Thus, whether IL-18 deficiency also contributes to human LrNK maturation and HCC suseptibility induced by early-life dysbiosis would also worthy of further investigation.

In summary, the present study demonstrate that early-life gut microbiota sustains functional maturation of LrNK cells through finely modulating liver microenvironment in a butyrate/IL-18 dependent manner. Our results reveal a crosstalk between microbiota and immunity in gut-liver axis and provide a potential intervention strategy after antibiotic exposure at early life.

## Methods

### Human samples
Human liver tissue included in this study were obtained from 10 hepatic (5 female and 5 male, aged: 26-65) hemangioma patients in Qilu hospital, Shandong University and Shandong Provincial Hospital from May 2022 to September 2022. All human tissues used in this study were approved by the Ethics Committee of Shandong University School of Basic Medical Sciences (ECSBMSSDU2019-1-41), and all patients provided informed written consent.

### Cell lines
The mouse cell line Yac-1 cells (BFN608006355) was donated by Shandong Academy of Medical Sciences and cultured in DMEM (Gibco) plus 10% FBS (Gibco) and 1% Penicillin-Streptomycin (solarbio). Cells were cultured at 37 °C in a constant temperature incubator with 5% $CO_2$.

### Bacterial preparation
*Clostridium butyricum* (*C. butyricum*) was supplied by the China General Microbiological Culture Collection Center and incubated in Reinforced Clostridial Medium under anaerobic conditions for 72 h at 37 °C. The bacteria were harvested by centrifugation (3000 *g* x 5 min) and resuspended in PBS to a final experimental concentration of $2 \times 10^9$ CFU/0.2 ml and gavage to early-Abx mice after weaning $2 \times 10^9$ CFU/0.2 ml 3 times a week until 8-week-old.

*Clostridium butyricum* Powder (trade name: Baolean, Qingdao Donghai Pharmaceutical Co., Ltd) was bought at ShuYu Civilian Pharmacy and gavage to early-Abx mice after weaning $2 \times 10^6$/0.2 ml per day until 8-week-old.

### Experimental animals
C57BL/6 mice (6–8 weeks of age) were purchased from Beijing Vital River Laboratory Animal Technology. IL-18Rα knockout mice were gifted by prof. Wei Wang from Sichuan University. CD45.1 mice were kindly provided by Dr. Xiaolong Liu (Center for Excellence in Molecular Cell Science, CAS). Mice were mated after 8 weeks of age and 4-5 male littermates from at least 2 dams were randomly assigned to different cages at post-weaning. All mice were maintained under specific pathogen-free conditions with a 12 h light, 12 h dark cycle and given free access to food and water. Experiments were carried out under the Shandong University Laboratory Animal Center's approval. Animal Ethics Number: ECSBMSSDU2020-2-86.

### Antibiotic treatment
Commensal microbes were depleted using antibiotics as previously reported[32]. Specifically, four kinds of antibiotics including ampicillin (1 g/l) (Solarbio), vancomycin (0.5 g/l) (Solarbio), neomycin sulfate (1 g/l) (Solarbio) and metronidazole (1 g/l) (Solarbio) were dissolved in sterile water and stored in 4 °C no more than a week before using. This antibiotic-contained water was supplied as drinking water to pregnant, breastfeeding mice and was changed every 3 days. Meanwhile, the offspring mice from pregnant C57BL/6 dams without antibiotic treatment were maintained in parallel as controls in each experiment. For early-Abx model, 8-week-old control or early-Abx male mice (18-22 g body weight) were subjected to HCC models, 16 S rRNA sequencing and microbiota metabolic or liver NK cell analysis (5-11 mice per group, born from 2-4 different dams and fed in 2-3 different cages).

### HCC mouse models
For AKT/Myc model, 8-week-old control or early-Abx male mice were hydrodynamically injected with the sleeping-beauty transposition system and AKT/Myc plasmids (2 ml physiological saline including 1 μg pCMV-SB100, 8 μg pT3-AKT and 16 μg pT3-cMyc). Six weeks later, the mice were sacrificed and liver tissues were photographed, and the phenotype and function of LrNK cells were analyzed. For depletion of both NK cells and ILC1 or cNK cells only, mice were intraperitoneally injected with 200 μg anti-NK1.1 (BioXcell) or 200 μg anti-asialo GM1 (WAKO) respectively, or the same volume of IgG (BioXcell) every three days for 4 weeks.

For STZ-HFD induced HCC model, a mouse model mimicking nonalcoholic fatty liver disease -related HCC[39], control or early-Abx male mice was induced by a single subcutaneous injection of 200 μg Streptozocin (Sigma) at 2 days after birth and feeding with HFD (cat#: D12492, Rodent diet with 60 kcal% fat) after 4 weeks of age. At month 5, the mice were sacrificed and tumor nodes in murine livers were photographed, and the phenotype and function of LrNK cells were analyzed.

The maximal tumor weight was not exceeded 10% weight of the chosen animal as stipulated by the Ethics Committee of Shandong University Laboratory Animal Center.

### Adoptive transfer model
CD3[-]NK1.1[+] cells were purified from 8-week-old control or early-Abx mice (CD45.2). Then $1 \times 10^6$ purified cells were transferred into CD45.1 mice. Twenty-four hours later, the recipient mice were intraperitoneally injected with 150 μg poly (I:C) (Sigma) to stimulate NK cell activation[62,63] and the function of CD45.2[+]CD49a[+]CD3[-]NK1.1[+] cells were analyzed 16 h later. On the other hand, CD3[-]NK1.1[+] cells or LrNK cells were puried from CD45.2 wild-type or IL-18Rα-deficient mice and separately transferred into control or early-Abx mice (CD45.1). Then, the function of CD45.2[+]CD49a[+]CD3[-]NK1.1[+] cells were analyzed in the recipient mice after injection of poly (I:C) for 16 h.

### Isolation of liver mononuclear cells (LMNCs), bone marrow mononuclear cells and cell subset purification
Briefly, the mouse livers were washed and passed through a 200-gauge stainless steel mesh. The single cells were washed, red blood cells were lysed and then the cell suspension were centrifuged over 40% Percoll gradient medium. For bone marrow mononuclear cells, bone marrow was obtained by flushing femurs and tibias and then RBC lysed and washed with PBS.

For purification of LrNK cells, Kupffer cells, IL-18Rα⁻ LrNK and IL-18Rα⁺LrNK cells, LMNCs were stained with specific antibodies and subjected to Moflo Astrios EQ.

For the isolation the liver mononuclear cells (LMNCs) from hepatic hemangioma patients, fresh tissues were washed with PBS, cut into small pieces and digested with collagenase typeII(1 mg/mL, Worthington) and DNaseI(0.01 mg/mL, ThermoFisher Scientific) in RPMI 1640 medium (Gibco) for 1 h at 37 °C. Then, lymphocytes were isolated with 30% Percoll density gradient and washed twice with PBS.

## Flow cytometry analysis

For cell surface staining, cell suspensions were incubated with the specific labeled antibodies for 30 min at 4 °C. For intracellular staining, freshly isolated cells were stimulated with PMA (50 ng/ml) (Sigma) and Ion (1 µg/ml) (Biolegend) for 2 h, or IL-12 (20 ng/ml) (Proteintech) and IL-15 (50 ng/ml) (Proteintech) for 16 h, then cultured with Brefeldin A (BFA) (Biolegend) at a final concentration of 10 µg/mL for 4 h. After surface staining, cells were fixed with intracellular fixation buffer for 20 min, then permeabilized with permeabilization buffer for 10 min. Intracellular staining was performed with antibodies diluted into permeabilization buffer. For CD107a staining, cells were incubated with CD107a antibody for 4 h. Flow cytometry was carried out on Cytoflex S (Beckman coulter) and analyzed by FlowJo 10.6.2 or CytExpert 2.3.0.

All antibodies for flow cytometry staining are shown in Supplementary Table 1.

## 16 S rRNA sequencing and microbiota metabolic analysis

Feces were immediately incubated on ice and rapidly transferred to −80 °C for storage after collection. The bacteria were tested by gene sequencing on the Illumina Hiseq platform with Majorbio (Shanghai, China). The V3-V4 region of the 16SrRNA genes was amplified using the universal primers (338 F: ACTCCTACGGGAGGCAGCA, 806 R: GGAC-TACHVGGGTWTCTAAT). The data were analyzed on the online platform of Majorbio Cloud Platform (www.majorbio.com).

For analysis of fecal microbiota metabolites, feces (-10 mg) were subjected to LC-MS by Shanghai Majorbio Bio-pharm Technology Co.,Ltd. Feces (-10 mg) or liver and spleen tissue (-100 mg) were subjected to a targeted GC-MS analysis to quantify short chain fatty acid levels.

## Isolation and culture of hepatocytes

Mice were anaesthetized and the portal vein was dropped with solution A (D-hanks without $Ca^{2+}$ and $Mg^{2+}$ containing 0.5 mM EGTA (Santa), 10 mM HEPEs (Coolaber) for 8 ml/min. When the liver was saturated, the inferior vena cava was opened and the mice were perfused with another 20 ml of solution A and 20 ml of solution B (Hanks' solution containing 10 mM HEPEs, 0.5 mg/ml Type IV Collagenase (Gibco)). Then the liver was taken out, washed, torn with tweezers, and digested with 10 ml mixture of solution A and solution B in an incubator at 37 °C for 20 min. The cell suspension was filtered, and centrifuged at 47 g. The cell pellet was resuspended in DMEM (Gbico) containing 10% fetal bovine serum (Gbico), 1% penicillin-streptomycin Liquid (Solarbio), 2 uM sodium pyruvate (Sigma), 0.4 µg/ml dexamethasone (Sigma), 14 U/L insulin (Solarbio and cultured for 3 h and then cultured in DMEM containing 10% fetal bovine serum, 1% penicillin-streptomycin, 2 µM sodium pyruvate, 0.04 µg/ml dexamethasone, 0.14 U/L insulin.

## Transwell co-culture assay

Co-culture assay was performed for LrNK cells with LMNCs, hepatocytes or Kupffer cells using Transwell system. Briefly, $1 \times 10^4$ purifed LrNK cells were plated in the transwell insert (upper compartment) and co-cultured with $1 \times 10^5$ LMNCs, hepatocytes or Kupffer cells at bottom of transwell (lower compartment) for 24 h. Poly(I:C), the GPCR

agonists AR-420626 (R&D), 4-CMTB (TOPSCIENCE) or MK-0354 (MCE) or butyrate (Sigma) were added at the indicated experiments.

For GPR109A knockdown, isolated hepatocytes or Kupffer cells ($1 \times 10^5$ cells) were grown overnight in 12-well plates and infected with negative control or GPR109A shRNA lentivirus (Shanghai Genechem Co.,Ltd.) and incubated for 72 h. Knockdown efficiency was verified by RT-qPCR and were used for the co-culture experiments.

## Cytotoxicity assay

CFSE-labelled YAC-1 cells were co-cultured with purified LrNK cells with IL-2 (100 U/mL) at the effector: target ratio of 5:1 at 37 °C in a 5% $CO_2$ incubator for 4 h. Then 7-AAD was stained, and FCM identified lysed cells as CFSE⁺7-AAD⁺.

## Seahorse analysis

$2 \times 10^6$ purified LrNK cells were pre-treated with IL-12 (20 ng/ml) and IL-15 (50 ng/ml) for 12 h, then seeded in a Seahorse Bioscience culture plate coated with Cell-Tak solution (Corning), and cultured in XF Base Medium Minimal DMEM medium (Agilent) with 25 mM glucose, 2 mM glutamine and 1 mM pyruvate in a non-$CO_2$ incubator for 1 h. Basal, maximal OCR and ATP production were measured by an XF96 Seahorse Extracellular Flux Analyzer (Agilent) following the manufacturer's instruction.

## Reverse transcription-quantitative PCR (RT-qPCR)

Total RNA was extracted from liver tissues or cells using TRIzol reagent. cDNA synthesis was done using Revert Aid First Strand cDNA Synthesis Kit and random primers according to the manufacturer instructions. PCR was carried out using SYBR® Green Real-Time-qPCR Master Mix. Primers pairs for target genes are shown in Supplementary Table 2.

## Statistical analysis

Statistical significance was determined using Prism 8. Student's $t$ tests (two-tailed unpaired) between two groups or one-way analysis of variance (ANOVA) with Tukey's multiple comparisons test were used to determine significance. The difference in overall survival was tested using log-rank tests. The results of 16 s rRNA were tested using Wilcoxon rank sum test and non-parametric factorial Kruskal−Wallis (KW) sum-rank test. Data are presented as mean ± SEM. Statistical significance was reported as $^*P < 0.05$; $^{**}P < 0.01$; $^{***}P < 0.001$; and ns, no significance.

## Reporting summary

Further information on research design is available in the Nature Portfolio Reporting Summary linked to this article.

## Data availability

Source data are provided with this paper. The 16 S rRNA sequencing datasets in this study have been deposited in the NCBI database with SRA, accession: PRJNA890468. The RNA-seq data are available under restricted access for NCBI database, access can be obtained by accession number PRJNA884315. The raw data are provided as a Source Data file. The authors declare that all data supporting the findings of this study are available within the paper and its Supplementary Information files or from the corresponding author upon reasonable request. Source data are provided with this paper.

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

## Acknowledgements

This work was supported by grants from the National Key Research and Development Program (No. 2022YFA1103402 (X.L.), 2021YFC2300603 (C.M.), 2018YFE0126500 (X.L.)), the National Science Foundation of China (No. 82171805 (X.L.), 81830017 (C.M.), 81970508 (X.L.)), Taishan Scholarship (No. tspd20181201(C.M.)), Shandong University multi-disciplinary research and innovation team of young scholars (2020QNQT001 (X.L.)), Major Basic Research Project of Shandong Natural Science Foundation (No. ZR2k02O2ZD12 (C.M.)). Thanks for the supporting from Translational Medicine Core Facility of Shandong University for consultation and instrument availability. Thanks to Professor Wang Wei of Sichuan University for presenting IL-18R α Knockout mice.

## Author contributions

X.L. and P.T. formulated the study concept. X.L. and P.T. designed the studies. P.T., W.Y. performed the experiments with assistance from X.G., T.W., S.T., R.X., R.S., Y.W., D.J., Y.Xu, Y.W. P.T. and X.L. analyzed the results. X.L., Z.W., C.L., L.G., and C.M. interpreted the results. P.T., X.L., L.G., and C.M. wrote and edited the manuscript.

## Competing interests

The authors declare no competing interests
