## [Peer Review File · Nature Communications]

Early life gut microbiota sustains liver-resident natural killer cells maturation via butyrate-IL-18 axisREVIEWER COMMENTS

Reviewer #1 (Remarks to the Author):

The authors Tian and collaborators propose in their manuscript that "Early-life gut microbiota sustains LrNK cell maturation via butyrate/IL18 axis". They aimed at deciphering the mechanisms linking gut microbiome and liver resident NK cells and highlighted that this process involves the gut bacteria – butyrate – GRP109A – IL18 pathway. The findings are of interest and explained in a comprehensive manner but relevance to humans still need to be determined or at least discussed. Other main reviewer's comments are listed below:

- A key finding is that microbial dysbiosis induced by early life antibiotic treatments in mice is a long lasting event that can persist in adulthood, which in turn will affect NK cells functionality in the liver. However, in humans, early microbiota perturbations, due to antibiotics, way of birth, feeding etc, which indeed might have long lasting effects on immune system or physiology, may not remain after the age of 3 years. Variables that might underlie the 8-weeks persistent alterations of the gut microbiome following antibiotic treatment in those mice are unclear.
- Antibiotic treatment is administered to both pregnant dams and pups. This may not allow disentangling the differential effect of in utero versus newborn/breastfeeding imprinting. Documenting the antibiotic cocktail's effects on the pregnant mice NK cells functionality might also be of interest.
- Features of mice might be better described (gender, weight). More importantly it is unclear whether each pup analysed is originating from one dam or whether several pups are born and fed from the same dam. Similarly the potential cage-effect is not addressed but one of these (cage or mother) could maybe explain the bimodal distribution detected in Fig3B?
- Within the metabolomic part, the authors mention that they performed GC-MS on fecal samples but data relate to cecal SCFA content (Fig4A). LC-MS is also performed on fecal samples, but differentially detected metabolites or pathways are not discussed. While the microbial-associated butyrate effect on immune cells activation via the GPR109A and IL18 might somehow be expected from literature, antibiotic treatment surely affects lots of non-SCFA metabolites (as mentioned in suppl FigS4A) but these are neither presented nor discussed.

More technical points are as follows:

- If similar mice are studied, Fig4C seems somehow irrelevant with Fig3C?
- Mice number in each experiment is rather low for statistical analyses (sometimes n=3 or n=4).
- Antibiotics are administered in drinking bottles, which usually leads to inter-individual variations. Those variations might be better shown in the figures (individual dots) and correlated with NK cells maturation and functions. Moreover antibiotic microbial depletion should also be evaluated not only based on relative abundance but on total abundance (by qPCR or flow cytometry), and also on diversity (here only richness is addressed in Fig3A).
- The primers used for 16SrRNA gene sequencing amplify the V3-V4 region of the 16S and not V4 only.

Reviewer #2 (Remarks to the Author):

The present study identifies a new crosstalk between gut microbiota and liver NK cells in which early-life gut microbiota sustains functional maturation of liver NK cells through modulating liver microenvironment in a butyrate/IL-18 dependent manner. The clinical relevance of this gut-liver axis is provided by the correlation observed in vivo with the progression of HCC. This is a well-designed and performed study, although, with the limitation of small number of samples (n=4). The manuscript is well written and all data are presented in systemic manner. The major concern raises from overly-optimistic statements of blocked liver NK cell maturation based on few phenotypic markers and functional response. Specific comments are provided below:

Major Comments

1. Introduction should better describe liver resident NK cell subsets, in particular their phenotype, effector functions in relation to the state-of arts for the development of the liver NK cells. In addition, for non-experts in mice model the corresponding human liver NK cell subsets should mentioned.
2. The hypothesis that gut microbiota sustain liver NK cells maturation, is confusing and little supported by the data. The authors clearly show several phenotypical and functional changes in the liver NK cells upon gut microbiota alterations, however, expression changes of markers CD27 or TIGIT associated with less mature NK cells along with the non-significant changes in the frequency do not necessary imply blocking in differentiation states of liver NK cells. Moreover, phenotypic comparison of liver NK cells with conventional blood and splenic NK cells in the context of NK cell maturation blocking is confusing since liver NK cells can develop separately from conventional NK cells (Cel. & Mol. Immu., 14; 2017). Thus, more specific characterization such as transcription factors, proliferation and rather comparison with bone marrow NK cells will be necessary to support the hypothesis of the maturation arrest of NK cells.
3. For more than 2 samples the statistical test ANOVA and not t-test should be used, example Fig. 2H, 2F-G etc.

Minor point:

1. What are the authors feeling about this model in human? This point should be discussed.
2. What is the rational for specific Poly:IC stimulation?
3. Explain better STZ-HFD-HCC model

Reviewer #3 (Remarks to the Author):

The manuscript of Tian et al describes the assessment of early-life antibiotic treatment on liver-resident NK cells, and reports a reduction in IrNK cell numbers and function in mice with early antibiotic treatment. The authors furthermore try to link this impairment of IrNK cells to a dysregulation of the butyrate/IL-18 axis. While studies investigating the impact of early antibiotic treatment on immune development are important, the significance of the current study is strongly reduced by a number of major limitations and concerns.

Overall, the authors describe the relevance of their data in the context of frequent treatment with antibiotics of infants, but in their experiments do not differentiate between treatment of

mothers with antibiotics or treatment of newborn mice. Furthermore, the studies are limited by a very small number of mice investigated in each arm, small differences between the groups, and a statistical analysis using Student's t-test (it is unclear if paired or unpaired, of if one- vs two-tailed) without adjusting for the many different comparisons performed. Finally, there are a number of concerns with the data presented in the figures as described below.

Figure 1: The number of mice per group is limited to 4, and differences between groups are very small (what is somehow hidden by showing MFI at values ranging from 10^3 to 10^4). Even if some results provide statistical significance (see comment above regarding concerns with statistical tests used and lack of adjustment for multiple comparisons), the biological significance of these findings is limited. An example is figure 1H, with minimal differences in lysis, but this limitation applies to all figures.

Figure 2: The authors state that impaired IrNK cell function due to early Ab treatment contributes to enhanced HCC progression, but figure 2H appears to show the opposite - mice with early Ab treatment (blue) survive longer than control mice (red).

Figure 3 and 4: the differences shown in these figures between control mice and Ab treated mice remain minimal, except for the expected differences in the microbiome, and the statistical analysis used is not adequate.

Figures 5-7. Unacceptably low number of mice are used for the experiments (3 per group?), and differences between groups are minimal/absent.

Responses to Reviewers

Thank you for your insightful comments and suggestions on our work. The manuscript has been carefully revised according to your comments. We believe that the revision has significantly improved our manuscript, and hope that you will also share the same opinion of ours. Our point-to-point responses to your specific comments are as follows.

To Reviewer#1:

1. The authors Tian and collaborators propose in their manuscript that "Early-life gut microbiota sustains LrNK cell maturation via butyrate/IL18 axis". They aimed at deciphering the mechanisms linking gut microbiome and liver resident NK cells and highlighted that this process involves the gut bacteria – butyrate – GRP109A – IL18 pathway. The findings are of interest and explained in a comprehensive manner but relevance to humans still need to be determined or at least discussed.

Reply: Thanks for the reviewer's comment and encouragement. As suggested, we analyzed the correlation of butyrate level in normal liver tissues from hepatic hemangioma patients with the maturation and function of LrNK cells. We found that, consistent with the mouse experimental data, LrNK cells from butyrate-high liver tissues had lower expression of CD27 (an immature marker for LrNK cells), but higher level of IFN- γ expression and CD107a mobilization than those cells from butyrate-low livers (Fig.1 for reviewers). These clinical data indicate that butyrate might also involve in the functional maturation of human LrNK cells. Furthermore, clinical studies reported that IL-18 has the ability to enhance the function of human liver NK cells¹, and human *IL-18* gene polymorphisms are associated with the risk and severity of HCC². However, whether early-life antibiotics treatment has correlation with liver butyrate level, IL-18

concentration and LrNK cell functional maturation would be further investigated in a large clinical cohort. The potential clinical relevance of our work was also discussed in the revised manuscript (page 20-21, line 436-441).

Fig. 1 for reviewers. (a) Grouping according to butyrate level in normal liver tissue from hepatic hemangioma patients. (b-c) Representative FACS plots (b) and bar graph (c) of the percentage of CD27, IFN- γ and CD107a in LrNK cells from patients with high levels of butyrate and those with low butyrate levels. Data are presented as means \pm SEM. * $p < 0.05$, *** $p < 0.001$, **** $p < 0.0001$.

2.A key finding is that microbial dysbiosis induced by early life antibiotic treatments in mice is a long-lasting event that can persist in adulthood, which in turn will affect NK cells functionality in the liver. However, in humans, early microbiota perturbations, due to antibiotics, way of birth, feeding etc, which indeed might have long lasting effects on immune system or physiology, may not remain after the age of 3 years. Variables that might underlie the 8-weeks persistent alterations of the gut microbiome following antibiotic treatment in those mice are unclear.

Reply: Thanks for your valuable comment. As the reviewer has correctly pointed out, many factors during early-life might perturb the gut microbiota,

which in turn have long-term consequences for immune homeostasis and disease susceptibility^{3, 4}. Here we found that mice with early-life antibiotics exposure showed altered microbial community composition in adulthood, compared to control mice. To minimize the effect of other factors on gut microbiota, littermates were randomly assigned to different cages at post-weaning. And all mice were housed in a specific pathogen-free facility with a 12h light, 12h dark cycle and given free access to food and water. In this way, variables, such as way of birth, feeding, growth environment, are basically identical between control and early-Abx mice, except the antibiotic treatment. The detailed description for mouse management has been added in Methods section in the revised manuscript (Page 21, line 453-458). However, we could not completely rule out that other factors contributing to the early dysbiosis might regulate LrNK cell maturation in human (Page 18, line 380-388).

3. Antibiotic treatment is administered to both pregnant dams and pups. This may not allow disentangling the differential effect of in utero versus newborn/breastfeeding imprinting. Documenting the antibiotic cocktail's effects on the pregnant mice NK cells functionality might also be of interest.

Reply: Thanks for the reviewer's helpful suggestion. This is a good point. Although several literatures established early-life dysbiosis model by feeding the pregnant and breastfeeding mice with antibiotic-contained water^{5, 6}, it is difficult to distinguish the differential effect in utero *versus* breastfeeding. Thus, we introduced utero-Abx and breastfeeding-Abx mouse model, in which mice were only exposed to antibiotic treatment in late pregnancy or during breastfeeding period, respectively. As shown in Supplementary Fig.4, antibiotic treatment either in utero (Supplementary Fig. 4a-c) or during breastfeeding period (Supplementary Fig. 4d-f) also impaired the maturation and function of LrNK cells.

As reviewer pointed out, the effect of antibiotic cocktail on NK cell homeostasis of the pregnant mice is also an interesting issue. Actually, there is still a lack of extensive data for the effects of antibiotics exposure on pregnant women. It was reported that pregnant women with the usage of antibiotics are potentially at increased risk of high level of glucose and insulin, as well as decreased alpha diversity of intestinal flora ⁷. Therefore, we investigated the effect of antibiotic treatment on LrNK cell maturation and function from the pregnant mice after delivery. Results showed that there were no differences in LrNK cell maturation and function between control and pregnant-Abx mice (Fig.2 for reviewers).

Fig. 2 for reviewers. Abx treatment had no effect on LrNK cell maturation and function. (a) Representative FACS plots and bar graph of the MFI of CD27, CD11b and KLRG1 in LrNK cells from control and pregnant-Abx mice (n=6 per group). (b) Representative FACS plots and bar graph of the MFI of IFN- γ and CD107a in LrNK cells from control and pregnant-Abx mice (n=6 per group). Data are presented as means \pm SEM. ns, no significance.

4. Features of mice might be better described (gender, weight). More importantly it is unclear whether each pup analysed is originating from one dam or whether several pups are born and fed from the same dam. Similarly the

potential cage-effect is not addressed but one of these (cage or mother) could maybe explain the bimodal distribution detected in Fig3B?

Reply: Thanks for the helpful comment. We're sorry for our negligence. For early-Abx model, 8-week-old, male control or early-Abx mice (18~22g body weight) were subjected to liver NK cell analysis or HCC models. Actually, early-life antibiotics exposure also impaired the maturation and function of LrNK in female mice (Fig. 3 for reviewers). That is, there is no gender difference for the effect of early-life microbiota on LrNK cell functional maturation. To keep the consistence in whole manuscript, male mice were chosen for all experiments. We have added this information in Material and Methods section in the revised manuscript (Page 21, line 453-458; page 22, line 465-472). The body weight of mice has been added in the revised (Supplementary Fig. 1a).

Fig. 3 for reviewers. Detection of the maturation and function of LrNK cells from female control or early-Abx mice. (a) Flow cytometry analysis of maturation marker levels in LrNK cells. (b) Flow cytometry analysis of IFN- γ production and CD107a mobilization in LrNK cells stimulated with PMA and Ion. Data are presented as means \pm SEM. *** $p < 0.001$, **** $p < 0.0001$.

We are very sorry for our incomplete representation. For early-Abx treatment mouse model, 2 pregnant C57BL/6 mice per cage were treated with antibiotics from approximately embryonic day 14 (E14) to the weaning of their

offspring. After weaning, 4~5 male or female littermates from at least 2 dams were randomly assigned to the same cage. Meanwhile, the offspring mice from pregnant C57BL/6 dams without antibiotic treatment were maintained in parallel as controls in each experiment. Five or more early-Abx or control mice, from at least two dams and two cages, were subjected for each experiment, including NK cell analysis, cell transfer or HCC model for. Thus, there is no mother- or cage-effect for the differences in LrNK cell functional maturation between control and early-Abx mice. Especially, in Fig. 3b, 16s rRNA sequencing was performed by using feces from 11 of control mice and 10 of early-Abx mice, which were born from 3-4 different dams and fed in 3 different cages. We also added this detailed information in Material and Methods section in the revised manuscript (Page 21, line 453-458; Page 22, line 465-472).

5. Within the metabolomic part, the authors mention that they performed GC-MS on fecal samples but data relate to cecal SCFA content (Fig4A). LC-MS is also performed on fecal samples, but differentially detected metabolites or pathways are not discussed. While the microbial-associated butyrate effect on immune cells activation via the GPR109A and IL18 might somehow be expected from literature, antibiotic treatment surely affects lots of non-SCFA metabolites (as mentioned in suppl FigS4A) but these are neither presented nor discussed.

Reply: We greatly appreciate the reviewer's helpful comment. As the reviewer has correctly pointed out, we did perform GC-MS on fecal samples to detect the level of SCFAs, we have made corrections for Fig.4a.

In addition, LC-MS revealed the difference in abundance and clustering of fecal metabolites between 8-week-old control and early-Abx mice. Further KEGG pathway showed the enrichment of differentially expressed metabolites in multiple metabolic pathways, including lipid metabolism, amino acid

metabolism and carbohydrate metabolism, all of which has been demonstrated to involve in regulating immune homeostasis^{8, 9, 10}. Although we further demonstrated that reduced butyrate level contributes to LrNK cell maturation and functional impairment, the probability that other metabolites might also involves in this inhibition could not be ruled out. As suggested, analysis of differentially expressed metabolites in feces between 8-week-old control and early-Abx mice has been added in supplementary Fig. 7c and discussed in the revised manuscript (page 10, line 199-205; page 19, line 409-414).

More technical points are as follows:

1. If similar mice are studied, Fig4C seems somehow irrelevant with Fig3C?

Reply: Thanks for the reviewer's comment. We performed all *in vitro* and *in vivo* experiments under the similar and strict condition as far as possible. 16S rRNA sequencing showed the significant alteration in the composition of gut flora in early-Abx mice. And, the difference in the mean relative abundance of top 15 predominant bacteria between control and early-Abx mice was shown in Fig. 3c. While, in Fig. 4c, we analyzed their difference in the abundance of butyrate-producing microbiome, such as *Faecalibacterium*, *Roseburia*, *Fusobacteria*, and *Eubacterium*, which are not included in top 15 abundant bacteria. Therefore, although similar mice were studied in Fig. 4c and Fig. 3c, they seem irrelevant.

2. Mice number in each experiment is rather low for statistical analyses (sometimes n=3 or n=4).

Reply: Thanks for the reviewer's kind suggestion. We have increased the number of mice for all experiments, with 5~11 mice for each experiment. We have included the mouse number for each experiment in the figure legends.

3. Antibiotics are administered in drinking bottles, which usually leads to inter-individual variations. Those variations might be better shown in the figures (individual dots) and correlated with NK cells maturation and functions. Moreover, antibiotic microbial depletion should also be evaluated not only based on relative abundance but on total abundance (by qPCR or flow cytometry), and also on diversity (here only richness is addressed in Fig3A).

Reply: Thanks for these insightful suggestions. As suggested by reviewer, all statistical results, including data for NK cell maturation and functions, were presented in dot plot graph, to apparently show inter-individual variations.

In addition, to further confirm the persistent alteration of gut microbiota caused by early life antibiotic depletion, total abundance of fecal microbiota was determined by blood agar plate culture of feces, as well as qPCR analysis for fecal bacterial DNA and total 16s rRNA. All results showed that adult early-Abx mice had significantly decreased abundance of gut microbiota. These results were showed in supplementary Fig. 6a-c and described in the revised manuscript (page 8, line172-175).

We are very sorry for our unclear representation. α -diversity and β -diversity of gut microbiota were shown in Fig. 3a and Fig. 3b respectively.

4. The primers used for 16SrRNA gene sequencing amplify the V3-V4 region of the 16S and not V4 only.

Reply: Many thanks for the reviewer's reminder. It's correct that the primers used for 16SrRNA gene sequencing amplify the V3-V4 region of the 16SrRNA genes. We have made correction in the revised manuscript (page24, line 524).

To Reviewer#2:

The present study identifies a new crosstalk between gut microbiota and liver NK cells in which early-life gut microbiota sustains functional maturation of liver NK cells through modulating liver microenvironment in a butyrate/IL-18 dependent manner. The clinical relevance of this gut-liver axis is provided by the correlation observed in vivo with the progression of HCC. This is a well-designed and performed study, although, with the limitation of small number of samples (n=4). The manuscript is well written and all data are presented in systemic manner. The major concern raises from overly-optimistic statements of blocked liver NK cell maturation based on few phenotypic markers and functional response. Specific comments are provided below:

Reply: Thanks for the reviewer's comment and encouragement. We have increased the number of mice for all experiments, with 5~11 mice for each experiment. Moreover, the alteration of LrNK cell maturation by early-life antibiotics exposure was demonstrated by analyzing the expression of phenotypic markers, transcription factors as well as functional responses (described in more details below). We hope that the revised manuscript could support the viewpoint that early-Abx treatment blocks the maturation of LrNK cells.

Major Comments

1. Introduction should better describe liver resident NK cell subsets, in particular their phenotype, effector functions in relation to the state-of arts for the development of the liver NK cells. In addition, for non-experts in mice model the corresponding human liver NK cell subsets should mentioned.

Reply: Thanks for the reviewer's helpful suggestion. We have added the

detailed description about the development, phenotype, and effector functions of both mouse and human liver resident NK cells in the Introduction section (page 2-3, line 30-33 and 38-44).

2. The hypothesis that gut microbiota sustain liver NK cells maturation, is confusing and little supported by the data. The authors clearly show several phenotypical and functional changes in the liver NK cells upon gut microbiota alterations, however, expression changes of markers CD27 or TIGIT associated with less mature NK cells along with the non-significant changes in the frequency do not necessary imply blocking in differentiation states of liver NK cells. Moreover, phenotypic comparison of liver NK cells with conventional blood and splenic NK cells in the context of NK cell maturation blocking is confusing since liver NK cells can develop separately from conventional NK cells (Cel. & Mol. Immu., 14; 2017). Thus, more specific characterization such as transcription factors, proliferation and rather comparison with bone marrow NK cells will be necessary to support the hypothesis of the maturation arrest of NK cells.

Reply: Thanks for these insightful comments. As the reviewer's correctly pointed out, we found that early-life antibiotics exposure upregulated the expression of CD27, an immature LrNK marker, and downregulated the levels of CD11b and KLRG1, two mature LrNK markers, while the percentage and number of LrNK cells, as well as the percentage of Lin⁻Sca-1⁺Mac-1⁺ (LSM) cells and CD49a⁺Lin⁻CD122⁺ LrNK progenitors, remained unaltered (supplementary Fig. 2e-f, page6, line117-123). These results suggest that early-life antibiotics treatment disrupts LrNK cell maturation, rather than their differentiation. Furthermore, to substantiate this viewpoint, we also detected the expression of transcription factors related with LrNK cell development as the reviewer suggested. As expected, RT-qPCR showed that compared with LrNK

cells from control mice, LrNK cells from 8-week-old early-Abx mice had reduced expressions of *Rora* and *Zfp683*, which have been reported as the important transcriptional factors for LrNK cells differentiation^{11, 12} (supplementary Fig.2b, page 6, line 112-115).

As suggested, we removed the results for spleen NK cells. In addition, we analyzed NK precursors in bone marrow in adult early-Abx and control mice. Flow cytometry analyses showed that there were no differences of the percentage or cell number of NKPs in bone marrow between control and early-Abx mice (Supplement Fig. 2g, page6, lin117-123).

In addition, in line with comparable numbers and percentage of LrNK cells, there was no significant difference in the proliferation and apoptosis of LrNK cells between early-Abx and control mice (Supplementary Fig. 1e-f, page 5, line 104-105).

3. For more than 2 samples the statistical test ANOVA and not t-test should be used, example Fig. 2H, 2F-G etc.

Reply : We are very sorry for our negligence. As suggested, ANOVA were used to determine significance for groups across variables or with multiple comparisons between groups. We have re-analyzed all related data and the description for statistical analysis method was corrected in the Method section (page 27, line 581-586).

Minor point:

1. What are the authors feeling about this model in human? This point should be discussed.

Reply: Thanks for the reviewer's insightful suggestion. Antibiotics exposure

was reported to potentially link with the increased incidence of tumors, including HCC¹³. Recently, it was also found that the decrease and functional impairment of LrNK cells caused by *ROR α* deficiency led to the acceleration of HCC development¹². Here we found that, LrNK cells from normal human liver tissues with high butyrate levels had lower expression of CD27 (an immature marker for LrNK cells), but higher level of IFN- γ expression and CD107a mobilization than those cells from butyrate-low livers (Fig.1 for reviewers), suggesting butyrate might also involve in the functional maturation of human LrNK cells. However, it would be also interesting to further investigate whether early-Abx treatment-induced LrNK maturation hindrance correlates with the increased risk of HCC. This point was also discussed in the revised manuscript (page18, line 380-388).

Fig. 1 for reviewers. (a) Grouping according to butyrate level in normal liver tissue from hepatic hemangioma patients. (b-c) Representative FACS plots (b) and bar graph (c) of the percentage of CD27, IFN- γ and CD107a in LrNK cells from patients with high levels of butyrate and those with low butyrate levels. Data are presented as means \pm SEM. * $p < 0.05$, *** $p < 0.001$, **** $p < 0.0001$.

2. What is the rationale for specific Poly(I:C) stimulation?

Reply: We're sorry for unclear description. The synthetic dsRNA polyinosinic-polycytidylic acid (poly I:C), a mimic of a common product of viral infections, has been reported to not only primes IL-18-mediated liver NK cell activation through directly acting on TLR3 on NK cells ¹, but also triggers Kupffer cells to produce IL-12, IL-18 and to express NKG2D ligand, Rae1 and then in turn indirectly promotes NK cell activation ¹⁴. Here we used Poly (I:C) stimulation to investigate the effect of microbiota-derived butyrate on LrNK cell functional maturation through modulating IL-18 production by Kupffer cells and hepatocytes. The description and reference have been added in the Materials and Methods section (page 23, line 490-492).

3. Explain better STZ-HFD-HCC model

Reply: Thanks for the reviewer's reminder. We have added the more detailed description for STZ-HFD-HCC model in the Materials and Methods section (page 22-23, line 482-487).

To Reviewer #3:

1. The manuscript of Tian et al describes the assessment of early-life antibiotic treatment on liver-resident NK cells, and reports a reduction in LrNK cell numbers and function in mice with early antibiotic treatment. The authors furthermore try to link this impairment of LrNK cells to a dysregulation of the butyrate/IL-18 axis. While studies investigating the impact of early antibiotic treatment on immune development are important, the significance of the current study is strongly reduced by a number of major limitations and concerns. Overall, the authors describe the relevance of their data in the context of frequent treatment with antibiotics of infants, but in their experiments do not differentiate

between treatment of mothers with antibiotics or treatment of newborn mice. Furthermore, the studies are limited by a very small number of mice investigated in each arm, small differences between the groups, and a statistical analysis using Student's t-test (it is unclear if paired or unpaired, of if one- vs two-tailed) without adjusting for the many different comparisons performed. Finally, there are a number of concerns with the data presented in the figures as described below.

Reply: We greatly appreciate the reviewer's agreement on the importance of early-life antibiotics treatment on immune development and the insightful comments. We do appreciate the helpful suggestions raised by reviewers. To further differentiate between treatment of mothers with antibiotics or treatment of newborn mice, we introduced utero-Abx and breastfeeding-Abx mouse model, in which mice were only exposed to antibiotic treatment in late pregnancy or during newborn period, respectively. As shown in Supplementary Fig.4, antibiotic treatment either in utero (Supplementary Fig. 4a-c) or during breastfeeding period (Supplementary Fig. 4d-f) also impaired the maturation and function of LrNK cells.

To further substantiate the conclusion of this work, we repeated all experiments for at least twice and increased the mouse number to 5~11 mice for each experiment. Moreover, the differences between the groups were re-analyzed. We hope that the revised version of manuscript has been improved.

We're very sorry for unclear description of statistical analysis. Two-tailed unpaired Student's t tests were used to determine the difference between two groups, while one-way ANOVA with Tukey's multiple comparisons test were used to determine significance for multiple groups. We have re-confirmed all statistical analysis data and described the statistical analysis method for each experiment in the figure legends. In addition, statistical analysis method has been described in more details in the Method section (page27, line 581-586).

The reviewer's concerns for the individual figure were replied below.

2. Figure 1: The number of mice per group is limited to 4, and differences between groups are very small (what is somehow hidden by showing MFI at values ranging from 10x3 to 10x4). Even if some results provide statistical significance (see comment above regarding concerns with statistical tests used and lack of adjustment for multiple comparisons), the biological significance of these findings is limited. An example is figure 1H, with minimal differences in lysis, but this limitation applies to all figures.

Reply: Thanks for the reviewer's helpful comments. As mentioned above, we repeated all experiments for at least twice and increased the mouse number to 5-11 mice for all experiments. Subsequently, the differences between the groups were re-analyzed in all figures. In the revised Figure 1, flow cytometry analysis showed the clear differences in the expression of maturation markers and effector molecules in LrNK cells, as well as their killing activity against target cells, between control and early-Abx mice. In addition, the positive percentages for those molecules expressed in LrNK cells were also analyzed and displayed in Supplementary Fig.2a and 3a. Results showed that both the MFI and percentage of maturation markers and effector molecules was significantly altered in LrNK cells from early-Abx mice, when compared to those in control mice. We assure that all these results were reproducible and the differences among groups were significant enough to support the viewpoint that early-life Abx treatment impairs LrNK functional maturation.

3. Figure 2: The authors state that impaired LrNK cell function due to early Ab treatment contributes to enhanced HCC progression, but figure 2H appears to show the opposite - mice with early Ab treatment (blue) survive longer than control mice (red).

Reply: We have double checked data in Figure 2H which show that mice with Abx treatment (blue) survive shorter than control mice (red). We are sorry for using confusing colors in original Figure 2H (red, blue, orange and pink). We have modified this panel in the revised manuscript. As shown in the new Figure 2h, although early-Abx mice (blue, straight line) developed more severe liver tumors than control mice (red, straight line), anti-NK1.1, but not anti-asialo GM1 (orange, dotted line for control mice; pink, dotted line for early-Abx mice), almost completely abrogated this effect. This result indicates that the impaired functional maturation of LrNK cells contributes to the promotion of early-Abx on HCC development.

4. Figure 3 and 4: the differences shown in these figures between control mice and Ab treated mice remain minimal, except for the expected differences in the microbiome, and the statistical analysis used is not adequate.

Reply: Thanks for the reviewer's insightful comments. We have repeated or re-analyzed the experiments in Figure 3 and Figure 4, and 5~11 mice or biological repeats for *in vitro* experiments have been used for each experiment. Moreover, both the MFI (Figure 3f-g, 4f-g) and positive percentage (Supplementary Fig. 6d-e, 7d-e) of maturation markers and effector molecules showed the significant alteration in LrNK cells from early-Abx mice, which was abrogated by co-housing treatment or butyrate supplementation. In addition, the statistical differences have been re-analyzed by using two-tailed unpaired Student's t tests for two groups' comparison or one-way ANOVA for multiple groups' comparison.

5. Figures 5-7. Unacceptably low number of mice are used for the experiments (3 per group?), and differences between groups are minimal/absent.

Reply: We appreciate for the reviewer's helpful comments. We have repeated the experiments in Figure 5-7, and 5~8 mice or biological repeats for *in vitro* experiments were used for each experiment. Moreover, both the MFI and positive percentage of maturation markers and effector molecules in LrNK cells has been analyzed. We believe that the differences among groups were significant enough to support the conclusion that butyrate / IL-18 axis promotes LrNK functional maturation by improving mitochondrial oxidative phosphorylation, and we hope that reviewers will share the same view with us.

Reference

1. Tu, Z., Hamalainen-Laanaya, H.K., Crispe, I.N. & Orloff, M.S. Synergy between TLR3 and IL-18 promotes IFN-gamma dependent TRAIL expression in human liver NK cells. *Cell Immunol* **271**, 286-291 (2011).
2. Teixeira, A.C. *et al.* Alleles and genotypes of polymorphisms of IL-18, TNF-alpha and IFN-gamma are associated with a higher risk and severity of hepatocellular carcinoma (HCC) in Brazil. *Hum Immunol* **74**, 1024-1029 (2013).
3. Cox, L.M. & Blaser, M.J. Antibiotics in early life and obesity. *Nat Rev Endocrinol* **11**, 182-190 (2015).
4. Brodin, P. Immune-microbe interactions early in life: A determinant of health and disease long term. *Science* **376(6596):945-950**. (2022).
5. Rakoff-Nahoum, S., Paglino, J., Eslami-Varzaneh, F., Edberg, S. & Medzhitov, R. Recognition of commensal microflora by toll-like receptors is required for intestinal homeostasis. *Cell* **118**, 229-241 (2004).
6. Li, F. *et al.* The microbiota maintain homeostasis of liver-resident gammadeltaT-17 cells in a lipid antigen/CD1d-dependent manner. *Nat Commun* **7**, 13839 (2017).
7. Su, Y. *et al.* Effect of exposure to antibiotics on the gut microbiome and biochemical indexes of pregnant women. *BMJ Open Diabetes Res Care* **9** (2021).

8. Hezaveh, K. *et al.* Tryptophan-derived microbial metabolites activate the aryl hydrocarbon receptor in tumor-associated macrophages to suppress anti-tumor immunity. *Immunity* **55**, 324-340 e328 (2022).
9. Kim, Y. *et al.* Dietary cellulose prevents gut inflammation by modulating lipid metabolism and gut microbiota. *Gut Microbes* **11**, 944-961 (2020).
10. Yoo, J.Y., Groer, M., Dutra, S.V.O., Sarkar, A. & McSkimming, D.I. Gut Microbiota and Immune System Interactions. *Microorganisms* **8** (2020).
11. Mackay LK, M.M., Kragten NA, Liao Y, Nota B, Seillet C, Zaid A, Man K, Preston S, Freestone D, Braun A, Wynne-Jones E, Behr FM, Stark R, Pellicci DG, Godfrey DI, Belz GT, Pellegrini M, Gebhardt T, Busslinger M, Shi W, Carbone FR, van Lier RA, Kallies A, van Gisbergen KP. Hobit and Blimp1 instruct a universal transcriptional program of tissue residency in lymphocytes. *Science* **352(6284)** (2016).
12. Song, J. *et al.* Requirement of RORalpha for maintenance and antitumor immunity of liver-resident natural killer cells/ILC1s. *Hepatology* (2021).
13. Yang, B. *et al.* Associations of antibiotic use with risk of primary liver cancer in the Clinical Practice Research Datalink. *Br J Cancer* **115**, 85-89 (2016).
14. Hou, X., Zhou, R., Wei, H., Sun, R. & Tian, Z. NKG2D-retinoic acid early inducible-1 recognition between natural killer cells and Kupffer cells in a novel murine natural killer cell-dependent fulminant hepatitis. *Hepatology* **49**, 940-949 (2009).

REVIEWERS' COMMENTS

Reviewer #2 (Remarks to the Author):

None

Reviewer #3 (Remarks to the Author):

The authors have addressed many of the concerns raised, in particular concerning sample sizes and statistical analyses. In the rebuttal, the authors provide data using human samples (Fig.1 for reviewers), but I was not able to find those data in the revised manuscript. The lack of data in the revised manuscript demonstrating a relevance of these finding in mice for humans remains a limitation of the study.

Responses to Reviewers

Thank you for your insightful comments and suggestions on our work. The manuscript has been carefully revised according to your comments. We believe that the revision has significantly improved our manuscript, and hope that you will also share the same opinion of ours. Our point-to-point responses to your specific comments are as follows.

Reviewer # 2 (Remarks to the Author):

None

Reply: We greatly appreciate the reviewer's careful evaluation of our manuscript.

Reviewer # 3 (Remarks to the Author):

The authors have addressed many of the concerns raised, in particular concerning sample sizes and statistical analyses. In the rebuttal, the authors provide data using human samples (Fig.1 for reviewers), but I was not able to find those data in the revised manuscript. The lack of data in the revised manuscript demonstrating a relevance of these finding in mice for humans remains a limitation of the study.

Reply: We greatly appreciate the helpful suggestion from the reviewer. As suggested, we added the data of human samples to figure 4 h-j including text (page12, line 243-249), human sample ethics (page22, line 474-480), the methods of isolation the liver mononuclear cells (LMNCs) (page 26, 560-565) figure legends (page 40, line 976-981).